# Fragment ion intensity prediction improves the identification rate of non-tryptic peptides in timsTOF

Charlotte Adams [1], Wassim Gabriel [2], Kris Laukens [1], Mario Picciani [2], Mathias Wilhelm [2,3], Wout Bittremieux [1] ✉ & Kurt Boonen [4,5] ✉

Immunopeptidomics is crucial for immunotherapy and vaccine development. Because the generation of immunopeptides from their parent proteins does not adhere to clear-cut rules, rather than being able to use known digestion patterns, every possible protein subsequence within human leukocyte antigen (HLA) class-specific length restrictions needs to be considered during sequence database searching. This leads to an inflation of the search space and results in lower spectrum annotation rates. Peptide-spectrum match (PSM) rescoring is a powerful enhancement of standard searching that boosts the spectrum annotation performance. We analyze 302,105 unique synthesized non-tryptic peptides from the ProteomeTools project on a timsTOF-Pro to generate a ground-truth dataset containing 93,227 MS/MS spectra of 74,847 unique peptides, that is used to fine-tune the deep learning-based fragment ion intensity prediction model Prosit. We demonstrate up to 3-fold improvement in the identification of immunopeptides, as well as increased detection of immunopeptides from low input samples.

The adaptive immune system can eradicate pathogen-infected and cancerous cells by recognizing peptides bound to major histocompatibility complex (MHC) molecules present on the cell surfaces. Even in the absence of infectious agents or cancerous transformation, the continuous yet dynamic process of peptide presentation informs the adaptive immune system about the health state of cells[1]. In immunopeptidomics, MHC-bound peptides—commonly termed immunopeptides—are isolated and characterized using mass spectrometry (MS). MS-based immunopeptidomics has been used to discover T cell targets against tumors, autoimmune diseases, and pathogens[2–5]. The identification of these targets is important for the development of immunotherapies, including the development of personalized vaccines and adoptive T cell transfers[6]. As even a single immunopeptide could elicit an immune response[7], potential targets can be based on a

single peptide-spectrum match (PSM). This underscores the importance of the specificity of PSM annotations.

Unfortunately, it remains challenging to identify immunopeptides from MS data. Because the generation of immunopeptides from their parent proteins lacks clear-cut rules, rather than being able to use known digestion patterns, every possible protein subsequence within human leukocyte antigen (HLA) class-specific length restrictions needs to be considered. As a result, there is a substantial inflation of the search space, leading to an increased false positive rate and a low peptide identification sensitivity[8]. In immunopeptidomics the search space is often expanded further by incorporating somatic mutations, pathogen genomes, and novel unannotated open reading frames (nuORFs) to be able to detect immunopeptides originating from these sources as well. A recent study highlights the significance of nuORFs as

[1]Department of Computer Science, University of Antwerp, Antwerp, Belgium. [2]Computational Mass Spectrometry, Technical University of Munich, 85354 Freising, Germany. [3]Munich Data Science Institute, Technical University of Munich, 85748 Garching, Germany. [4]Department of Biomedical Sciences, University of Antwerp, Antwerp, Belgium. [5]Sustainable Health Department, Flemish Institute for Technological Research (VITO), Antwerp, Belgium. ✉e-mail: wout.bittremieux@uantwerpen.be; kurt.boonen@uantwerpen.be

an underexplored source of MHC-I-presented, tumor-specific peptides that hold potential as targets for immunotherapy[9].

To minimize false positives and improve identification rates, PSM rescoring can be used. This involves post-processing results from an unfiltered database search using tools such as Percolator[10], to use multiple PSM features to distinguish between correct and incorrect PSMs. Recently, driven by powerful prediction tools, there has been significant interest in using additional features for PSM rescoring. One example is using MS/MS spectrum prediction tools to generate spectral features based on the similarity between experimental and predicted fragment ion intensities. This approach is especially relevant for immunopeptidomics, where the use of specialized fragment ion intensity prediction tools has yielded promising results[11–13]. For example, the use of Prosit led to a more than two-fold increase in the identification of HLA ligands[12].

A timsTOF mass spectrometer (Bruker) combines two stages of trapped ion mobility spectrometry (TIMS) with a quadrupole and a high-resolution time-of-flight (TOF) mass analyzer. This configuration introduces an additional dimension, the collisional cross section, that can separate isobaric peptides. During a single TIMS scan, multiple precursors can be selected as a function of ion mobility, while the first TIMS accumulates ions for the next TIMS scan. This scan mode, termed parallel accumulation-serial fragmentation (PASEF), increases MS/MS rates more than ten-fold without any loss in sensitivity[14].

In the context of immunopeptidomics, it is critical to use highly sensitive instrumentation due to the relatively low abundance of immunopeptides. A timsTOF-based approach has been shown to significantly increase HLA peptide identifications compared to immunopeptidomics using an Orbitrap mass spectrometer[15]. Furthermore, it has been demonstrated that optimization of the timsTOF acquisition method improves HLA peptide identification rates[16]. Moreover, a recent study has revealed that MS/MS spectra from timsTOF instruments exhibit more reproducibility at low abundances compared to MS/MS spectra from Orbitrap instruments[17]. Notably, when analyzing a hybrid proteome mixture using different instruments, substantial differences in fragment ion intensities were observed between timsTOF Pro and Orbitrap QE HF-X mass spectrometers[18]. While PSM rescoring has been proven to be highly effective for immunopeptides measured on an Orbitrap[12], the considerable dissimilarity in MS/MS spectra produced by timsTOF and Orbitrap instruments necessitates the development of fragment ion intensity prediction models that are optimized for predicting timsTOF data.

In this study, we measure 302,105 unique synthesized non-tryptic peptides from the ProteomeTools project[19] on a timsTOF-Pro to fine-tune the existing Prosit model[12]. The integration of fragment ion intensity predictions into PSM rescoring of search results significantly improves the identification rate of HLA peptides measured on a timsTOF. In addition, we rescore timsTOF data from low-input samples and successfully identify immunopeptides derived from nuORFs.

## Results

### Measuring non-tryptic peptides on a timsTOF

The ProteomeTools project is a large-scale effort in which peptides were synthesized and analyzed. Initially it contained measurements of 330,000 synthetic tryptic peptides covering essentially all canonical human proteins[19]. Subsequently, the project expanded to include post-translational modifications[20] and non-tryptic peptides[12]. This valuable dataset was used to train the deep neural network, Prosit, for the prediction of retention time (RT) and fragment ion intensity[21]. However, all measurements conducted to train previous Prosit models were performed on Orbitrap and ion trap instruments.

The considerable dissimilarity in MS/MS spectra generated by timsTOF and Orbitrap instruments for the same peptide (Fig. 1, Supplementary Fig. 1) underscores the need to develop fragment ion intensity prediction models optimized for timsTOF data. To address this, we measured 302,105 unique synthesized non-tryptic peptides from the ProteomeTools project[12]. Our measurements encompassed a range of collision energies from 20.81 eV to 69.77 eV. Consequently, we compiled a dataset consisting of 93,227 non-tryptic MS/MS spectra from 74,847 unique peptides, complemented by 184,552 previously published tryptic MS/MS spectra from 138,201 unique synthetic tryptic peptides[22]. This extensive dataset, comprising a total of 277,779 MS/MS spectra and 213,048 unique peptides, serves as a unique training dataset for the development of machine learning tools tailored to timsTOF instruments (Supplementary Fig. 2).

### Optimized Prosit model improves prediction accuracy of tryptic and non-tryptic peptide timsTOF MS/MS spectra

To optimize the Prosit fragment ion intensity prediction model towards timsTOF instruments, we fine-tuned the HCD Prosit 2020 model using the 277,779 MS/MS spectra compiled in this study, split into training, validation, and test sets (Fig. 2a). The HCD Prosit 2020 model was selected because higher-energy collisional dissociation (HCD) is a non-resonant activation technique like the collision induced dissociation conducted in a TOF instrument[23]. The HCD

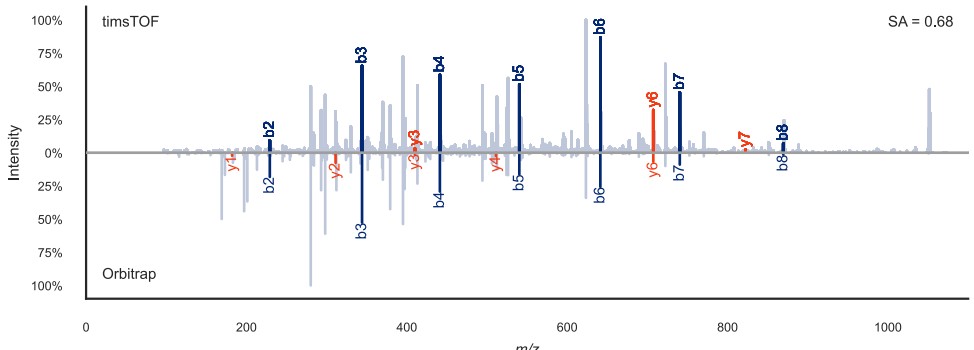

**Fig. 1 | Mirror plot illustrating the spectral variability between timsTOF and Orbitrap instruments.** Mirror plot of the singly charged non-tryptic synthetic ProteomeTools[19] peptide VEDPVTVEY measured on a timsTOF (top; mzspec:PXD043844:HLAI_p2_97_178_p2-D1_S1-D1_1_6866.mgf:index:2341:VEDPVTVEY/1) and on an Orbitrap (bottom; mzspec:PXD021013:02446d_GD1-TUM_HLA_133_01_01-3xHCD-1h-R4:scan: 32024:VEDPVTVEY/1) instrument. The spectral similarity measured by the normalized spectral contrast angle based on annotated

fragments between the two spectra is 0.68. This illustrates how different timsTOF MS/MS spectra can look compared to Orbitrap data. This randomly chosen peptide was measured several times on the Orbitrap, after which the spectrum with the highest similarity to all other Orbitrap spectra for this peptide was selected (the medoid spectrum). In the timsTOF data the displayed MS/MS spectrum was the only measurement of this peptide. SA normalized spectral contrast angle.

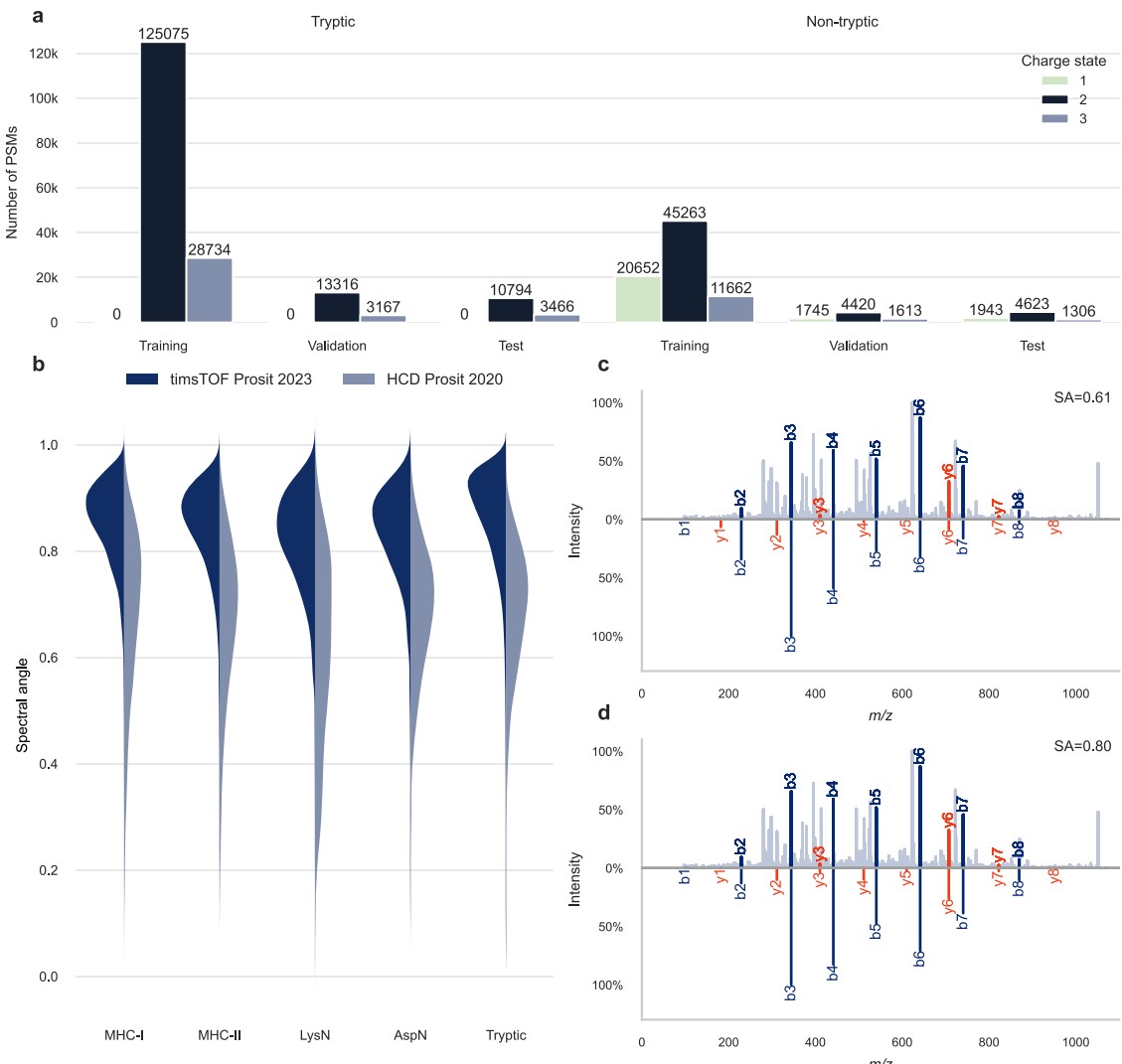

**Fig. 2 | Deep learning framework Prosit for tryptic and non-tryptic peptide fragment ion intensity prediction. a** The 277,779 MS/MS spectra from tryptic and non-tryptic peptides measured on timsTOF instruments were split into training, validation, and test sets and used to fine-tune Prosit. **b** Violin plots comparing the prediction accuracy of the timsTOF Prosit 2023 model against the previously published HCD Prosit 2020 model[12] for non-tryptic (MHC-I, MHC-II, LysN, and AspN) and tryptic peptides. **c** Mirror plot of the randomly chosen singly charged non-tryptic synthetic peptide VEDPVTVEY measured on a timsTOF (top; mzspec:PXD043844:HLAI_p2_97_178_p2-D1_S1-

D1_1_6866.mgf:index:2341:VEDPVTVEY/1) and the predicted spectrum for this peptide at the aligned collision energy with the HCD Prosit 2020 model (bottom). **d** Mirror plot of the same measurement (top; mzspec:PXD043844:HLAI_p2_97_178_p2-D1_S1-D1_1_6866.mgf:index:2341:VEDPVTVEY/1) and the predicted spectrum for this peptide at the aligned collision energy with the timsTOF Prosit 2023 model (bottom). Fragment ions are labeled in blue and red for b and y ions, respectively. The mirror plot was generated using spectrum_utils version 0.4.1[53]. SA normalized spectral contrast angle. Source data are provided as a source data file.

Prosit 2020 model was originally trained on approximately 30 million MS/MS spectra, consisting of 9 million MS/MS spectra of non-tryptic peptides[12] and 21 million previously published tryptic MS/MS spectra[19,21]). The comparison between the HCD Prosit 2020 model and the timsTOF Prosit 2023 model (Fig. 2b–d) reveals a substantial improvement in normalized spectral contrast angle (SA) between predicted and experimental timsTOF MS/MS spectra for non-tryptic peptides (SA ≥ 0.9 for 26.3% of spectra, compared to 2.4% with HCD Prosit 2020) and for tryptic peptides (SA ≥ 0.9 for 42.1% of spectra, compared to 0.2% with HCD Prosit 2020). The timsTOF Prosit 2023 model demonstrates consistent performance across different precursor charges, with only a minor decrease of the performance for peptides with charge state 3 compared to charge state 1 and 2 (one-way ANOVA followed by Tukey's post hoc test; Supplementary Fig. 3a). This indicates that it is more challenging to predict accurate fragment ion intensities for more complex spectra with higher precursor charges. We also observed a moderate influence of

the peptide length on the predicted fragment ion intensities, indicating that accurate fragment ion intensity prediction is more challenging for longer peptides (Pearson correlation coefficient of −0.38; Supplementary Fig. 3b). The timsTOF Prosit 2023 model demonstrates consistent performance across different collision energies, with only a minor influence on the predicted fragment ion intensities (Pearson correlation coefficient of 0.22 for the collision energy; Supplementary Fig. 3c). We observed a small bias in function of C- and N-terminal amino acids, specifically, the performance is higher for peptides with an arginine or a lysine at the C-terminus (one-way ANOVA followed by Dunnett's post hoc test, Supplementary Fig. 4a, b). This is likely a result of the large proportion of tryptic peptides in the training data (Supplementary Fig. 2a).

It is important to note that the applied collision energy has a profound impact on the information content of the obtained MS/MS spectra[24] (Supplementary Fig. 1). To optimize the transfer learning, the collision energies of the training, validation, and test set were

calibrated according to the collision energies the HCD Prosit 2020 model would expect. To achieve this, robust linear models were trained on the training set, stratified by precursor charge state and tryptic status, and then applied on the training, validation, and test sets (see Methods).

## PSM rescoring boosts immunopeptide identification on timsTOF

We hypothesized that integrating fragment ion intensity predictions into PSM rescoring of search results would improve the identification rate of HLA peptides measured on a timsTOF, similar to what was previously observed for tryptic and non-tryptic peptides measured on other instruments[12]. To investigate this, we reanalyzed data from a recently published benchmarking study on timsTOF-based immunopeptidomics for tumor antigen discovery[15]. The study compared timsTOF-based immunopeptidomics to immunopeptidomics using Orbitrap technology and demonstrated a significant increase in the identification of immunopeptides from various benign and malignant primary samples of solid tissue and hematological origin.

In this analysis, the dataset was reprocessed with MaxQuant and all proposed PSMs were PSM rescored by integrating Prosit's fragment ion intensity predictions and retention time predictions, using Oktoberfest[25]. We compared results from MaxQuant + Prosit + Percolator vs MaxQuant + Percolator, respectively, the feature sets can be found in Supplementary Tables 1 and 2. This allowed us to evaluate PSM rescoring of timsTOF data using the timsTOF 2023 model and PSM rescoring of Orbitrap data using the CID 2020 model for HLA-I and HCD 2020 model for HLA-II (Fig. 3a–d). PSM rescoring of the Orbitrap data resulted in on average 2.2-fold more unique HLA-I peptides and 1.4-fold more unique HLA-II peptides. PSM rescoring timsTOF data resulted in comparable results, with on average 2.7-fold more unique HLA-I peptides and 1.8-fold more unique HLA-II peptides. Because the current Prosit models were not trained on peptides containing free cysteine side chains or other amino acid modifications that may be identified on immunopeptides, 8% of potential target PSMs (of which 99% would not survive the posterior error probability <0.01 filter) were lost because they could not be rescored.

To evaluate the effect of the fragment ion intensity prediction model, PSM rescoring was performed on the timsTOF data with all three models. To isolate the benefit of fragment ion intensity information during rescoring, RT prediction-based features were excluded during this analysis (Supplementary Table 12). As expected, PSM rescoring with the timsTOF Prosit 2023 model consistently resulted in a higher gain of identifications compared to PSM rescoring with the HCD Prosit 2020 and the CID Prosit 2020 model (Fig. 3e, f, Supplementary Fig. 5e, f). A possible explanation as to why PSM rescoring with the HCD Prosit 2020 model consistently resulted in a higher increase compared to PSM rescoring with the CID Prosit 2020 model, is that both HCD and timsTOF have a beam-type fragmentation and are thus more similar compared to CID[23]. The evaluation of the effect of the different fragment ion intensity prediction models on PSM rescoring of the Orbitrap data achieved similar results, with the CID Prosit 2020 model consistently resulting in a higher gain of identifications in the HLA-I dataset and the HCD Prosit 2020 model consistently resulting in a higher gain of identifications in the HLA-II dataset (Supplementary Fig. 6).

## PSM rescoring boosts the identification rate of relevant immunopeptides in low-input samples

To enable the detection of rare and clinically relevant antigens from a limited cell input, Phulphagar et al.[26] developed a high-throughput single-shot MS-based immunopeptidomics workflow using the timsTOF single-cell proteomics system (SCP). This workflow was applied to sample inputs ranging from 1 million to 40 million A-375 cell equivalents, a melanoma cell line which expresses the following HLA-I genes: A*01:01, A*02:02, B*57:01, B*44:03, C*16:02, and C*06:02.

This experiment was reprocessed to evaluate how PSM rescoring with the timsTOF Prosit 2023 model would perform on data from low-input samples. Individual spectrum peak files were searched against a compiled database consisting of the human reference proteome, common laboratory contaminants, curated small open reading frames (ORFs), and novel unannotated ORFs (nuORFs) supported by ribosomal profiling[9]. All proposed PSMs by MaxQuant were subsequently rescored using Oktoberfest. The results showed an average increase in identified HLA-I ligands across different input sizes, ranging from 1.3-fold at 1 million cell equivalents to 1.9-fold at 40 million cell equivalents (Fig. 4a, Supplementary Fig. 7a). Interestingly, we observed a consistent median spectral angle around 0.85 across all cell equivalents, supporting the finding that MS/MS spectra from timsTOF instruments are reproducible at low abundances[17] (Supplementary Fig. 7b).

To validate the peptide identifications obtained through PSM rescoring, we employed Gibbs clustering[27] on the gained, shared, and lost peptides separately. We then compared the cluster motifs with the known binding motifs of the HLA alleles expressed by the cells. The selection of motifs shown in Fig. 4b was based on the cluster with the highest Kullback–Leibler distance. The Kullback–Leibler distance provides a measure of similarity between clusters, thus identifying the cluster that differs the most from the other clusters found. Notably, we observed that the clusters with the highest Kullback–Leibler distances to the other clusters among the shared and gained peptides exhibited a striking resemblance to the motif of A*01:01. Conversely, the motifs of the clusters of the lost peptides did not correspond to any of the motifs of the HLA types present in the cell line (Fig. 4b, Supplementary Fig. 8). The motifs of the other clusters based on the gained and shared peptides were consistent with other HLA alleles present in the cell, namely A*02:02, B*44:03, and B*57:01 (Supplementary Fig. 8).

To further validate the peptide identifications obtained through PSM rescoring, we assessed the predicted binding affinity of the gained, shared, and lost peptides. Using thresholds provided by NetMHCpan[28] for weak binders and strong binders, we found that 88% of peptides gained after PSM rescoring were weak binders of at least one of the HLA types present in the cell, with 80% being a strong binder (Fig. 4c). For the shared peptides this was 89% and 85%, and for the lost peptides this was 44% and 24%, respectively. This implies that 56% of the peptides lost after PSM rescoring were predicted to not bind any of the HLA molecules present in the cell.

Among the identified immunopeptides, a subset of 2509 peptides (3%) originated from nuORF source proteins (Fig. 4d). Recent studies have provided evidence that peptides derived from noncanonical proteins can be displayed on HLA-I molecules[29,30]. These nuORFs may arise from transcripts that are currently annotated as non-protein coding, including the 5' and 3' untranslated regions, overlapping yet out-of-frame alternative ORFs in annotated protein-coding genes, long noncoding RNAs, or pseudogenes[9]. HLA peptides derived from noncanonical proteins can expand the repertoire of potential immunotherapy targets in cancer. Notably, we did not observe significant changes in the ratio of nuORFs after PSM rescoring, indicating a robust FDR control in proteogenomics. In addition, more than twice as many nuORF source proteins were identified after PSM rescoring the 40 million cell equivalent samples, which are of great interest. Furthermore, we examined the binding affinity of peptides originating from nuORFs and found that 90% of peptides can be considered a weak binder to at least one of the HLA types present in the cell, with 81% being a strong binder. This suggests that these peptides are actually presented by the cell.

## PSM rescoring improves the identification rate of samples cleaved with different proteases

To further evaluate how PSM rescoring performs when different proteases are used, we performed a reanalysis of samples cleaved with

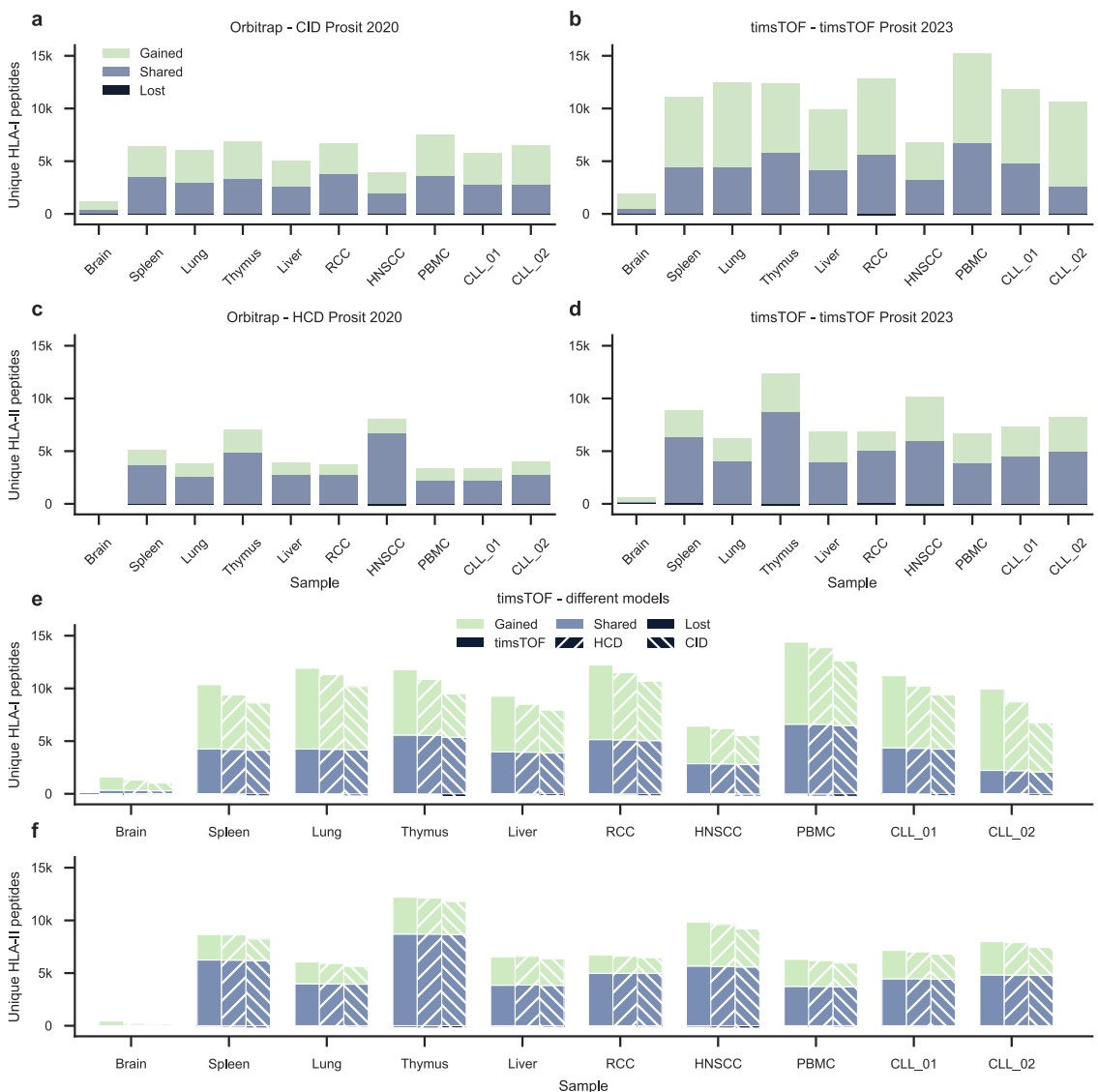

**Fig. 3 | Gained, shared, and lost peptide identifications for different sample types to compare PSM rescoring on Orbitrap data with PSM rescoring on timsTOF data.** In general PSM rescoring was able to boost the confidence in peptide identifications, retaining true PSMs, gaining PSMs, and losing only a small number of previously incorrect PSMs. **a** On average PSM rescoring of HLA-I Orbitrap data with the CID Prosit 2020 model resulted in a 2.2-fold increase. **b** On average PSM rescoring of HLA-I timsTOF data with the timsTOF Prosit 2023 model resulted in a 2.7-fold increase. **c** On average PSM rescoring of HLA-II Orbitrap data with the HCD Prosit 2020 model resulted in a 1.4-fold increase. **d** On average PSM rescoring of HLA-II timsTOF data with the timsTOF Prosit 2023 model resulted in a 1.8-fold increase. **e** To evaluate the effect of the fragment ion intensity prediction model on PSM rescoring, the RT prediction-based features were excluded. On average PSM rescoring of HLA-I timsTOF data with the timsTOF Prosit 2023 model resulted in a 2.8-fold increase. On average PSM rescoring of HLA-I timsTOF data with the HCD Prosit 2020 model resulted in a 2.5-fold increase. On average PSM rescoring of HLA-I timsTOF data with the CID Prosit 2020 model resulted in a 2.2-fold increase. **f** To evaluate the effect of the fragment ion intensity prediction model on PSM rescoring, the RT prediction-based features were excluded. On average PSM rescoring of HLA-II timsTOF data with the timsTOF Prosit 2023 model resulted in a 1.5-fold increase. On average PSM rescoring of HLA-II timsTOF data with the HCD Prosit 2020 model resulted in a 1.5-fold increase. On average PSM rescoring of HLA-II timsTOF data with the CID Prosit 2020 model resulted in a 1.5-fold increase. RCC renal cell carcinoma, HNSCC head and neck squamous-cell carcinoma, PBMC peripheral blood mononuclear cell, CLL chronic lymphocytic leukemia. Source data are provided as a source data file.

either trypsin, AspN, or GluC[31]. These three proteases have distinct cleavage sites, with trypsin cleaving at the C-terminal side of lysine and arginine residues[32], AspN mainly cleaving at the N-terminal side of aspartic acid residues[33], and GluC mainly cleaving at the C-terminal side of glutamic acid residues[33].

PSM rescoring of the samples cleaved with AspN, GluC, and trypsin resulted in 1.5-fold, 1.7-fold, and 1.4-fold increases in unique identified peptides, respectively (Fig. 5a). The performance of the timsTOF Prosit 2023 model was stable across all proteases, with a median spectral angle of 0.77, 0.78, and 0.80 for samples cleaved with AspN, GluC, and trypsin, respectively (Fig. 5b).

## Discussion

The identification of immunopeptides is critical for the advancement of vaccine and immunotherapy development. Previous studies have shown that using fragment ion intensity predictions in rescoring can greatly increase the identification rate of HLA ligands[11–13]. In this study, we established an extensive dataset consisting of 277,779 MS/MS spectra from synthetic non-tryptic and tryptic peptides measured on a timsTOF instrument. While not all synthesized peptides that are theoretically present could be identified, which could be due to both factors related to the acquisition method and the bioinformatics analyses, a valuable ground truth dataset was constructed that served as

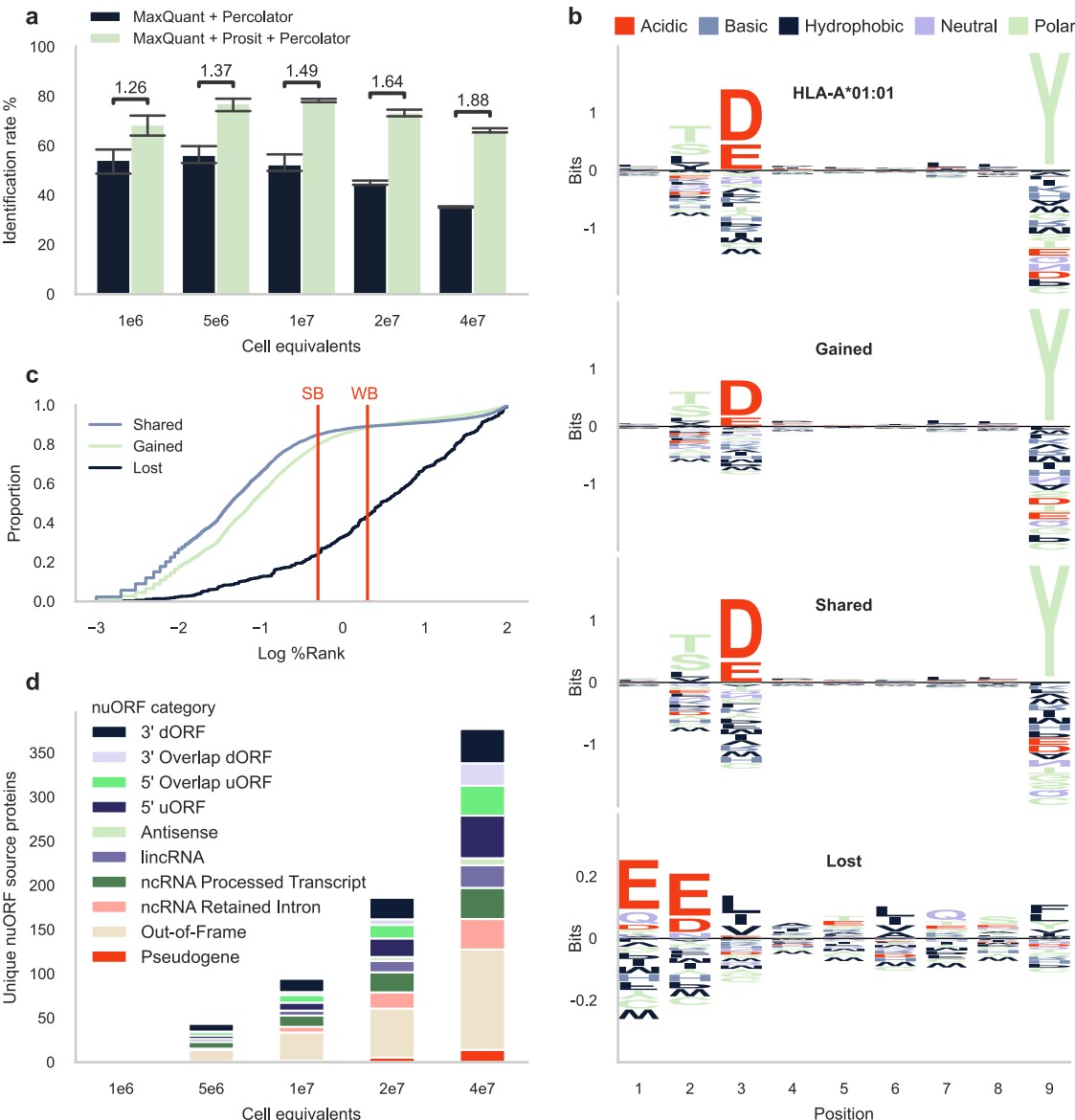

**Fig. 4 | PSM rescoring timsTOF SCP immunopeptidomics data. a** Bar chart illustrating the mean identification rates of 1 million to 40 million 375 cell equivalents. The height of each bar corresponds to the mean identification rate, while the whiskers indicate the standard deviation. Each sample was measured in technical triplicate (four technical replicates for the 40 million sample). Above each bar the fold change is shown between MaxQuant + Percolator and MaxQuant + Prosit + Percolator. **b** Peptide motif plots of 1406 unique peptides confidently identified to be present in the cell line expressing allele A*01:01[52], and peptide motif plots of 16,641 unique gained peptides, 13,208 unique shared peptides, and 447 unique lost peptides resulting from PSM rescoring of the 1 million to 40 million A-375 cell equivalents. Amino acids are colored according to their physicochemical properties (red acidic, blue basic, black hydrophobic, purple neutral, and green polar amino acids). **c** Cumulative distribution function (CDF) plot illustrating the distribution of the shared, gained, and lost peptides across the log10 percentile rank (%Rank) calculated with NetMHCpan version 4.1[28]. The cut-off values for strong binders (SB) and weak binders (WB) are indicated in red. **d** Mean unique nuORF source proteins contributing to the HLA-I immunopeptidome of 1 million to 40 million A-375 cell equivalents[9]. Source data are provided as a source data file.

the foundation for training the timsTOF Prosit 2023 model. By employing this model for PSM rescoring of MaxQuant results from immunopeptides measured on a timsTOF, we achieved a nearly 3-fold increase in the identification of HLA-I peptides. In addition, we demonstrated the effectiveness of our model for PSM rescoring of low sample inputs measured using a timsTOF SCP instrument, resulting in improved identification rates. Importantly, the immunopeptides identified after rescoring are likely to be HLA binders, as supported by the motif analysis and binding affinity assessment, providing an orthogonal validation of our method. Moreover, PSM rescoring led to an almost 2-fold increase in the identification of unique nuORF source

proteins, which hold the potential to serve as valuable targets for immunotherapy[29,30].

In addition to the analysis of immunopeptidomics data, our model holds promise for numerous other biological and biomedical applications. One such area is deep proteome sequencing, where multiple proteases are used to enhance proteomic coverage[34], particularly in regions with suboptimal trypsin cleavage sites, such as membrane-spanning domains and splice junctions. Our model can effectively enhance the confidence of peptide identifications in such studies, enabling valuable insights into alternative splicing and facilitating a comprehensive exploration of its impact on the proteome.

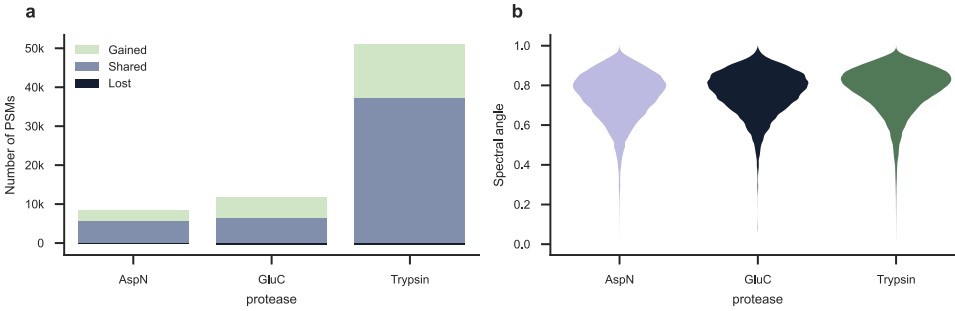

**Fig. 5 | PSM rescoring of samples cleaved with either AspN, GluC, or trypsin.** In general, PSM rescoring was able to boost the annotation rate for samples from all three enzymes, retaining true PSMs, gaining PSMs, and losing only a small number of previously incorrect PSMs. **a** PSM rescoring of the sample cleaved with AspN resulted in a 1.5-fold increase. PSM rescoring of the sample cleaved with GluC resulted in a 1.7-fold increase. PSM rescoring of the sample cleaved with trypsin resulted in a 1.4-fold increase. **b** The performance of the timsTOF Prosit 2023 model was assessed using violin plots depicting the normalized spectral contrast angle between predicted and observed spectra. The median spectral angle observed for AspN, GluC, and trypsin respectively, was 0.77, 0.78, and 0.80. Source data are provided as a source data file.

To ensure the timsTOF Prosit 2023 model's strong predictive capabilities towards immunopeptides, we generated MS/MS spectra from synthesized non-tryptic peptides to compile the training data. This enabled the model to generalize over different peptide types, whereas machine learning models that are solely trained on tryptic data often fail to do so, for example, by exhibiting strong biases based on the presence of C-terminal arginine or lysine residues. Although our model still exhibits a slight performance improvement for tryptic peptides, likely due to a larger number of such training samples, this did not preclude us from achieving strong performance for non-tryptic peptides. A potential limitation could be that while our current model only predicts fragment ion intensities for canonical b and y ions, non-tryptic peptides exhibit distinct MS/MS characteristics compared to tryptic peptides, often displaying strong internal ion series and neutral losses. However, as PSM rescoring using Prosit has demonstrated robustness against the presence of a large number of neutral loss or internal ion series[12], we do not expect this to be overly detrimental. A limitation to take into consideration is that our model is not able to predict fragment ion intensities for peptides containing free cysteine side chains or PTMs that were not seen during training.

It is important to note that the applied collision energy has a profound impact on the information content of the obtained MS/MS spectra[24]. Thus, collision energy calibration is needed for accurate fragment ion intensity predictions. The impact of collision energy in the timsTOF instrument is a bit more complicated compared to its impact in the Orbitrap. During IMS, ions are subjected to a series of collisions. This kinetic energy can be transferred to internal energy, similarly to what takes place during the activation of ions in collision-induced dissociation. Because IMS energizes the peptides significantly, the use of lower collision energies is advised[24]. Similarly to what has been observed for retention time alignment[35], we expect a benefit from collision energy alignment to account for the run-to-run fluctuations. Therefore, the collision energy calibration is implemented in Oktoberfest for PSM rescoring[25].

Another potential application of our timsTOF ground-truth dataset could be to develop a CCS prediction model for non-tryptic peptides. Similar to fragment ion intensity and RT predictions, CCS predictions can be used as features during PSM rescoring[36]. Some CCS prediction models already exist[22,37–39], including a recent model trained on tryptic peptides, phosphopeptides, and immunopeptides[38].

Although currently the timsTOF Prosit 2023 model is dependent on MaxQuant, as it relies on the search engine to sum the MS/MS scans, in the future, it will be further extended to support other search engines as well and become search engine agnostic, similar to how Oktoberfest has recently extended applicability of Prosit

Orbitrap predictions beyond MaxQuant. Based on previous studies[11–13] we expect an improvement of the identification rate when applied to other search engines as well. The timsTOF Prosit 2023 model is available on Koina (Prosit_2023_intensity_timsTOF, https://koina.proteomicsdb.org) and can be used via Oktoberfest.

## Methods

### Data acquisition

**Synthetic non-tryptic peptides data acquisition.** Within the ProteomeTools project, 302,105 unique non-tryptic peptides were synthesized, comprising 168,688 HLA class I, 73,464 HLA class II, 31,744 AspN, and 31,435 LysN sequences. For detailed information on the peptide origins, please refer to the original publication by Wilhelm et al.[12]. Peptide pools for synthesis and measurement contained roughly 1000 peptides each. Near-isobaric peptides (±10 p.p.m.) were distributed across different pools of similar length to avoid ambiguous masses in pools wherever possible. Ten microliters of the stock solution were transferred to a 96-well plate and spiked with two retention time standards (Pierce Retention Time Standard and PROCAL[40]) at 100 fmol per injection. An equimolar amount of approximately 50 fmol of each peptide was injected into an Evosep One HPLC system (Evosep) coupled to a hybrid TIMS-quadrupole TOF mass spectrometer (Bruker Daltonik timsTOF Pro) via a nano-electrospray ion source (Bruker Daltonik Captive Spray). The 100 samples per day (SPD) method was used. The Endurance Column 15 cm × 150 μm ID, 1.9 μm beads (EV1106, Evosep) was connected to a Captive Spray emitter (ZDV) with a diameter 20 μm (1865710, Bruker) (both from Bruker Daltonics).

The timsTOF Pro was calibrated according to the manufacturer's guidelines. The source parameters were: capillary voltage 1500 V, dry gas 3.0 l/min, and dry temp 180 °C. The temperature of the ion transfer capillary was set to 180 °C. The column was kept at 40 °C. The data-dependent Parallel Accumulation-Serial Fragmentation (PASEF) method was used to select precursor ions for fragmentation with 1 TIMS-MS scan and 10 PASEF MS/MS scans, as described by Meier et al.[14]. The TIMS-MS survey scan was acquired between 0.70 and 1.70 Vs/cm$^2$ and 100–1700 $m/z$ with a ramp time of 100 ms. The $m/z$ and ion mobility information was used to select precursors with charges ranging from 1 to 3. No polygon was used for precursor ion selection. Dynamic exclusion was used to avoid re-selecting of precursors that reached a target value of 20,000 a.u. The timsTOF Pro was controlled by the OtofControl 6.0 software (Bruker Daltonik GmbH). The collision energy was increased as a function of decreasing ion mobility (ranging from 0.76–1.68 Vs/cm$^2$), starting from 20 eV to 70 eV.

**Synthetic tryptic peptides data acquisition.** The "proteotypic" synthetic peptide set from ProteomeTools[19], covering confidently and

frequently identified proteins (124,875 peptides covering 15,855 human annotated genes), was obtained by Meier et al.[22]. The data was downloaded from the PRIDE repository with identifier PXD019086.

As per Meier et al., LC–MS was performed on an EASY-nLC 1200 (Thermo Fisher Scientific) system coupled to timsTOF Pro mass spectrometer (Bruker Daltonik, Germany) via a nano-electrospray ion source (Bruker Daltonik Captive Spray). Approximately 200 ng of peptides were separated on an in-house 45 cm × 75 μm reversed-phase column at a flow rate of 300 nL/min in an oven compartment heated to 60 °C. The column was packed in-house with 1.9 μm C18 beads (Dr. Maisch Reprosil-Pur AQ, Germany) up to the laser-pulled electrospray emitter tip. Mobile phases A and B were water and 80%/20% ACN/water (v/v), respectively, and both buffered with 0.1% formic acid (v/v). The pooled synthetic peptides were analyzed with a gradient starting from 5% B to 30% B in 35 min, followed by linear increases to 60% B and 95% in 2.5 min each before washing and re-equilibration for a total of 5 min.

The timsTOF Pro was operated in data-dependent PASEF[41] mode with 1 survey TIMS-MS and 10 PASEF MS/MS scans per acquisition cycle. They analyzed an ion mobility range from 1/K0 = 1.51 to 0.6 Vs/cm$^2$ using equal ion accumulation and ramp time in the dual TIMS analyzer of 100 ms each. Suitable precursor ions for MS/MS analysis were isolated in a window of 2 Th for $m/z$ < 700 and 3 Th for $m/z$ > 700 by rapidly switching the quadrupole position in sync with the elution of precursors from the TIMS device. The collision energy was lowered stepwise as a function of increasing ion mobility, starting from 52 eV for 0–19% of the TIMS ramp time, 47 eV for 19–38%, 42 eV for 38–57%, 37 eV for 57–76%, and 32 eV until the end. The $m/z$ and ion mobility information was used to exclude singly charged precursor ions with a polygon filter mask. Dynamic exclusion was used to avoid re-sequencing of precursors that reached a target value of 20,000 a.u.

### Preparation of the training data
The raw Bruker data from synthetic peptides from ProteomeTools[19] were analyzed with MaxQuant version 2.1.2.0[42]. Individual spectrum peak files were searched against pool-specific databases[43]. Default parameters were used, unless mentioned otherwise: carbamidomethylated cysteine was specified as a fixed modification and methionine oxidation as a variable modification. The minimal sequence length was set to 7 and the maximum sequence length was set to the maximum length of peptides in the pool. For timsTOF data the precursor mass tolerance was set to 10 ppm and the fragment mass tolerance was set to 40 ppm. For Orbitrap data the precursor mass tolerance was set to 4.5 ppm and the fragment mass tolerance was set to 20 ppm. Only the top PSM was used per spectrum and PSMs were filtered at a 0.01 posterior error probability (PEP). Only peptides expected in the pool, including full-length and N-terminally truncated peptides, were selected. All PSMs, even for the same peptide, with an Andromeda score ≥70 were included.

Unprocessed spectra were extracted from the raw Bruker files with OpenTIMS[44], using the precursorID from the accumulatedMsmsScans.txt and the frameID from the pasefMsmsScans.txt MaxQuant output files. Frame-level scans were summed based on the scan number from msms.txt with MasterSpectrum version 1.1[45]. The b and y ions were annotated for fragment charges ranging from 1 to 3.

The data were split into three distinct sets with each peptide and subsequence of a peptide only included in one of the three: training (80%, 153,809 tryptic PSMs and 77,577 non-tryptic PSMs), validation (10%, 16,483 tryptic PSMs and 7,778 non-tryptic PSMs), and test (10%, 14,262 tryptic PSMs and 7,872 non-tryptic PSMs). For each PSM in the training set, MS/MS spectra were predicted with the HCD Prosit 2020 model across collision energies ranging from 5 to 45 eV. The SA was calculated between the observed spectra and the predicted spectra, and the collision energy corresponding to the top-scoring predicted

spectra was selected as the optimal collision energy. This process was performed separately for each peptide type (tryptic, non-tryptic) and precursor charge state (1–3). A robust linear model was trained using RANSAC regression in scikit-learn version 1.2.2[46] to predict the difference between the reported collision energy and the optimal collision energy, based on the peptide mass.

To calibrate the validation and test set, the collision energy difference was predicted for each peptide mass, and this difference was applied to obtain the aligned collision energy. The models used for the collision energy calibration are available on the MassIVE repository (MSV000092456).

### Prosit 2023 model training
The HCD Prosit 2020 model[12] was fine-tuned using the training set. To control for overfitting, the validation set was used with early stopping, employing a patience of 5 epochs. The test set was used after the model was fully trained to evaluate its generalization and potential biases.

The model architecture remained unchanged, and the normalized spectral contrast loss[21] was used as a loss function. We used the Adam optimizer[47] with a cyclic learning rate algorithm[48]. During training, the learning rate cycled between a constant lower limit of 0.00001 and an upper limit of 0.0002 which is continuously scaled by a factor of 0.95 with the "triangular" mode. The model was trained with a batch size of 2000 on an Nvidia V100 GPU. The model improved significantly in predicting fragment ion intensity during the initial epochs, as depicted in Supplementary Fig. 9, and converged at epoch 28 with a median SA of 0.86.

### Statistical analysis
To evaluate the consistency of the timsTOF Prosit 2023 model across different peptide lengths and collision energies, a Pearson correlation was performed, resulting in Pearson correlation coefficients of −0.38 and 0.22, respectively. In addition, we investigated the influence of the charge state on the fragment ion intensities with a one-way ANOVA, resulting in a $p$-value of 6.97E−247. Subsequently, Tukey's post hoc test was used to assess the differences between charge states 1 and 2 ($p$-value = 0.96), 1 and 3 ($p$-value = 5.56E−12), and 2 and 3 ($p$-value = 5.56E−12). To evaluate the effect of N- and C-terminal amino acids on the model's performance, a one-way ANOVA was used, resulting in $p$-values 1.85E−27 and 3.68E−60, respectively. Then, Dunnett's post hoc test was used to compare peptides with a C-terminal arginine or lysine to all peptides ($p$-value = 4.95E−09). All statistical tests were performed on the test set.

### General PSM rescoring pipeline
Before PSM rescoring, all spectrum peak files were searched using MaxQuant version 2.0.3.1 with default parameters unless specified otherwise: carbamidomethylated cysteine was specified as a fixed modification and methionine oxidation as a variable modification. For timsTOF data the precursor mass tolerance was set to 10 ppm and the fragment mass tolerance was set to 40 ppm. For Orbitrap data the precursor mass tolerance was set to 4.5 ppm and the fragment mass tolerance was set to 20 ppm. The minimum peptide length was set to 8 amino acids and the maximum peptide length depended on the HLA class, with a length set to 16 amino acids for HLA-I and 30 amino acids for HLA-II. Specific settings for the individual datasets are detailed below.

The unfiltered search results, including decoy PSMs, were used as an input for the PSM rescoring with Oktoberfest version 0.6.0[25]. In brief, unprocessed MS/MS spectra corresponding to the identifications were extracted from the raw Bruker files and the b and y ions were annotated at fragment charges 1 up to 4. Collision energies were calibrated by predicting the top 100 scoring PSMs for each charge state with a collision energy range between 5 and 45 eV. The SA was calculated between the observed spectra and the predicted spectra

and the optimal collision energy was determined by selecting the collision energy corresponding to the top-scoring predicted spectra. A robust linear model was trained using RANSAC regression in scikit-learn version 1.2.2[46] to predict the difference between the reported collision energy and the optimal collision energy, based on the peptide mass. Subsequently, this model was used to determine the optimal collision energy. Both retention time and fragment ion intensities were predicted and features were generated (Supplementary Table 1) to add to Percolator[49], which was used for the PSM and peptide FDR estimation.

### Application of Prosit and PSM rescoring to external datasets

**Re-analysis of the comparison dataset.** To compare the PSM rescoring performance on Orbitrap versus timsTOF data, we utilized a comparison dataset comprising both HLA-I and HLA-II peptides measured on an Orbitrap and on a timsTOF. For detailed information on data acquisition, please refer to the original publication by Gravel et al.[15]. In brief, 10 samples were measured in technical triplicate (two technical replicates for the HNSCC sample) on the Orbitrap Fusion Lumos mass spectrometer (Thermo Fisher Scientific, Waltham, USA) and on the timsTOF Pro (Bruker Daltonik, Germany). The fragmentation methods used for the Orbitrap instrument were resonance-type collision-induced dissociation (CID) at a normalized collision energy of 35% for HLA-I peptides and higher-energy collisional dissociation (HCD) at a normalized energy of 30% for HLA-II peptides. The data was downloaded from the PRIDE repository with identifier PXD038782.

Individual spectrum peak files were searched against a database containing 20,598 human UniProt entries downloaded from https://www.ebi.ac.uk/reference_proteomes/ on 23/03/2023[43]. Carbamidomethylated cysteine was not included as a fixed modification, because cysteine was not carbamidomethylated during sample processing. Before PSM rescoring all PSMs containing free cysteine side chains were removed. The Orbitrap data was searched with a precursor tolerance of 20 ppm and the timsTOF data with a precursor tolerance of 40 ppm. To perform PSM rescoring on the Orbitrap data we employed the 2020 CID Prosit model with a collision energy set to 35 for HLA-I peptides, and the 2020 HCD Prosit model with collision energy set to 30 for the HLA-II peptides. For timsTOF data, PSM rescoring was performed using the timsTOF Prosit 2023 model with the calibrated collision energies.

To evaluate the effect of the fragment ion intensity prediction model, PSM rescoring was performed on the timsTOF data with the different models. To isolate the benefit of fragment ion intensity information during rescoring, RT prediction-based features were excluded during this analysis. The features abs_rt_diff, lda_scores, pred_RT, and iRT were removed from the list of features detailed in Supplementary Table 2.

**Re-analysis of an immunopeptidomics dataset measured on timsTOF SCP.** To investigate whether low input samples benefit from PSM rescoring, we rescored a timsTOF SCP dataset. For detailed information on data acquisition, please refer to the original publication by Phulphagar et al.[26]. In brief, HLA-I peptides were directly enriched from 1 million to 40 million A-375 cell equivalents by single shot injections on timsTOF SCP. Each sample was measured in technical triplicate (four technical replicates for the 40 million sample). Individual spectrum peak files were searched against a compiled database comprised of the human reference proteome Gencode 34 (ftp.ebi.ac.uk/pub/databases/gencode/Gencode_human/release_34) with 47,429 non redundant protein-coding transcript biotypes mapped to the human reference genome GRCh38, 602 common laboratory contaminants, 2043 curated small ORFs (lncRNA and upstream ORFs), 237,427 novel unannotated ORFs (nuORFs) supported by ribosomal profiling nuORF DB v1.037, for a total of 287,501 entries[9]. The data were downloaded from the PRIDE repository with identifier PXD040740.

To validate the peptide identifications acquired through PSM rescoring, gained, shared, and lost peptides were clustered with GibbsCluster version 2.0[27] with parameters for MHC class I ligands of length 8–13. Based on the Kullback–Leibler distance in function of the number of clusters, the optimal number of motifs in the data was selected. For each motif the position-specific scoring matrix was extracted and put into Seq2logo version 2.0[50] to get the position-specific frequency matrix of the Kullback–Leibler logos. All logos were visualized using the Python package Logomaker version 0.8[51]. The logos from gained, shared, and lost peptides were plotted next to the logos of the HLA-types present in the cell line to which they had the lowest Kullback–Leibler distance. For the HLA motif, peptide lists of the large monoallelic HLA class I cell line study by Sarkizova and Klaeger et al.[52] were used.

For each peptide we calculated the binding affinity to every HLA allele present in the cell line, using NetMHCpan version 4.1[28]. For each peptide the best, i.e. lowest, percentile rank value was retained. A percentile rank cutoff of 2 was used for weak binders and 0.5 for strong binders[28].

**Re-analysis of a dataset with multiple proteases.** To investigate whether samples digested with trypsin, GluC, or AspN benefit from PSM rescoring, we rescored a timsTOF Pro dataset that was digested using different proteases. For detailed information on data acquisition, please refer to the original publication by Fossati et al.[31]. In brief, for spectral library generation 500 ng for each fraction were acquired using DDA PASEF. The number of missed cleavages was fixed to 2, using cysteine carbamidomethylation as fixed modification, and N-terminal acetylation and methionine oxidation as variable modifications. Individual spectrum peak files were searched against a combined human-Mtb database encompassing the *Mycobacterium Tuberculosis* proteome (4,081 entries, downloaded from Uniprot on 12/02/2021) and Homo Sapiens proteome (20,397 entries, downloaded on 07/01/2021). The data were downloaded from the PRIDE repository with identifier PXD025671.

### Reporting summary
Further information on research design is available in the Nature Portfolio Reporting Summary linked to this article.

## Data availability
The MS datasets are available via the PRIDE and MassIVE repositories with the identifier MSV000092456 [https://massive.ucsd.edu/ProteoSAFe/dataset.jsp?task=357750c7e94a4ec0924a5df9a0e70705] (non-tryptic timsTOF dataset, also accessible with PXD043844), PXD019086 (tryptic timsTOF dataset; reanalysis available on MSV000092462 [https://massive.ucsd.edu/ProteoSAFe/dataset.jsp?task=14823bb1cb6e4508a180d8d5b6b179eb]), PXD038782 (comparison dataset; reanalysis available on MSV000092461 [https://massive.ucsd.edu/ProteoSAFe/dataset.jsp?task=b64af9dc5bb2430a8bf9e0d11d977b76]), PXD025671 (multiple proteases dataset; reanalysis available on MSV000093954 [https://massive.ucsd.edu/ProteoSAFe/dataset.jsp?task=06e103c793aa4faeb467efa59d2200c1]), and MSV000091456 [https://massive.ucsd.edu/ProteoSAFe/dataset.jsp?task=3a576bb329b14c709adfc2efad54b70b] (SCP dataset; reanalysis available on RMSV000000693.1 [https://massive.ucsd.edu/ProteoSAFe/reanalysis.jsp?task=bfe0766db1464a17add1acfda374ac68]). All protein databases used in this study are deposited alongside the result files. Source data are provided with this paper.

## Code availability
Source code and scripts are available on GitHub at https://github.com/wilhelm-lab/koina, https://github.com/wilhelm-lab/oktoberfest, and https://github.com/adamscharlotte/timsTOF-immunopeptide-prediction.

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

## Acknowledgements

We would like to acknowledge OpenMS and the Chan Zuckerberg Initiative for awarding C.A the OpenMS Computational Mass Spectrometry Fellowship [513040]. We thank Erik Fransen for his valuable insights on which statistical tests to use. This research was in part funded by the Research Foundation Flanders (FWO) [G070722N] (C.A., K.B.), European Union's Horizon 2020 Program under Grant Agreement 823839 [H2020-INFRAIA-2018-1; EPIC-XS] (W.G.), an ERC Starting Grant [101077037] (W.G.), and the German Federal Ministry of Education and Research (BMBF) [Grant No 031L0305A] (M.P.).

## Author contributions

K.B., W.B., and M.W. jointly supervised the research, providing guidance throughout the project. K.B. measured the non-tryptic synthetic peptides on the timsTOF-Pro, C.A. analyzed the data. C.A and W.G. trained and evaluated the Prosit timsTOF 2023 model. C.A. and W.G. conceptualized timsTOF processing in Oktoberfest. C.A. and M.P. implemented timsTOF support in Oktoberfest. C.A. wrote the manuscript with help of W.G., K.B., W.B., M.W., and K.L. All authors reviewed and approved the manuscript.

## Competing interests

K.L. is a co-founder and shareholder of ImmuneWatch BV: an immunoinformatics company. M.W. is a co-founder and shareholder of MSAID GmbH and OmicScouts GmbH, with no operational role in both companies. K.B. is a co-founder and shareholder of ImmuneSpec BV. The remaining authors declare no competing interests.
