## [Peer Review File · Nature Communications]

Reviewers' Comments:

Reviewer #1:

Remarks to the Author:

The authors describe a fine-tuned version of the ProSIT model for MS2 fragment ion intensity and precursor retention time prediction. The model is an iteration of the HCD ProSIT 2020 model, fine-tuned on newly acquired and publicly available timsTOF MS2 fragmentation spectra of non-tryptic peptides. The model is tested on a publicly available benchmarking dataset and on newly acquired samples. The authors adequately demonstrate that timsTOF fragmentation spectra are considerably different from Orbitrap spectra, and that using the model as part of rescoring MaxQuant search results is demonstrated to increase identification rates.

I found the manuscript and methods easy to read and understand for the most part. I was convinced that the newly trained model is a valid and important contribution to the immunopeptidomics community. However, several points need to be addressed before the manuscript should be considered for publication. Beyond minor revisions to clarify findings, text and figures, there are two major points. First, the central theme of the manuscript is that it is fragment intensity prediction that is driving the performance of the model-based rescoring pipeline. However, the specific contribution of fragment ion intensity to the improvement after rescoring is never assessed independently of all other features contributing to rescoring. Second, the results tend to be overinterpreted and conclusions should be toned down to match the presented findings.

Major revision points

1) Rescoring in general vs. fragment intensity prediction

The manuscript proposes an improvement in identification rate due to fragment ion intensity prediction. The improvement in identification rate is clearly shown in Figure 3 and 4, yet it is not shown that this is caused by fragment intensity prediction. What is shown is that the rescoring improves the identification rate, and the rescoring includes fragment intensity prediction, retention time prediction, and all the other features which are passed as inputs to Percolator. Thus, the contribution of fragment intensity prediction has not been assessed. The contribution of each factor (beyond the standard Percolator input) compared to a suitable reference (in this case MaxQuant and MaxQuant + the same rescoring pipeline presently used without any additional predictions) as well as the contribution of combinations of factors needs to be investigated before attributing factor-specific claims. Thus, Figure 3 needs to be extended to visualize the effect of the individual one- and two-factor models. Otherwise, the wording throughout the manuscript needs to be adapted regarding this claim, e.g. "Rescoring improves identification rate..." instead of "Fragment ion intensity prediction improves...".

2) Result overinterpretation

Interpretation of results is exaggerated and should be down-toned to match the described findings. The following points should be addressed.

2a) Accurate prediction

The second paragraph in the result section is headlined with "model allows accurate prediction". It should be rather rephrased as "model improves prediction accuracy" unless it can be shown that the accuracy achieved is in-line with the variation observed for technical replicates.

2b) TimsTOF vs. Orbitrap

The third paragraph states that "rescoring boosts identification on timsTOF compared to Orbitrap". The findings, however, show that "rescoring boosts identification on timsTOFs as well as Orbitrap". The difference is 2.8 vs. 2.5 and 1.7 vs. 1.4 fold for timsTOF vs. Orbitrap rescoring. It is not shown that this average improvement by 0.3 is significant. These results show the power of rescoring and that this power can be utilized for both MS platforms, but it isn't clear that there is a substantial difference

between the platforms. A helpful comparison here could be showing how the using the timsTOF model on timsTOF data improves the identification rate compared to using the orbitrap model on timsTOF data. This would very directly show the improvement when using an instrument-specific model.

2c) Novel neo-epitopes

The fourth results section claims that rescoring "reveals novel neo-epitopes". However, the description describes a proteogenomics search for the identification of peptides potentially derived from a non-coding region. These should be referred to as non-canonical peptides, not neoepitopes. Whole-exome sequencing is necessary to define somatic mutations (neoantigens) which give rise to neoepitopes, and this is absent. While the term neoepitopes is misused in various publications, its definition in the context of cancer is clear. See, for instance, Chong et al., NatCom 2020 (PMID 32157095).

Furthermore, the attribute "novel" is not substantiated. There is neither a comparison with previous publications (if the identified peptide candidates have been described elsewhere) nor is there an experimental verification that the peptides identified with statistical uncertainty are supported by mRNA evidence or synthetic spike-in experiments standard in the field (e.g. Skipper et al., Int J Cancer 1999, PMID 10417764).

From the second paragraph of this section: "Consequently, the benefit of rescoring is expected to be even higher for low input samples. To validate this assumption, we performed a reanalysis on this timsTOF SCP dataset using the TOF Prosit 2023 model." – The results of the titration series, which directly follow this passage, show the opposite of this assumption. It is shown that the largest fold-increase is achieved for the highest-input sample, and there appears to be a trend of lessening fold-increase with lessening sample input. Further, the results described in this section indicate a lower fold-increase than was observed when rescoring the benchmarking study, counter to what we are led to expect. Of course, it is not a problem to have data that doesn't support a specific hypothesis, but if it actually is a hypothesis you intended to test then please add some interpretation as to why the results are counter to expectations. If it isn't a hypothesis, but just an introductory sentence to the analysis, it might be better to phrase it as an evaluation of the rescoring on low-input data, rather than validation of an assumption of how the rescoring will perform.

Lastly, since rescoring was shown to increase the total number of identifications, it is expected that there will be an absolute higher number of non-canonical peptides identified as well. This effect could be reported in any peptide population investigated and is not specific to non-canonical peptides, so the title of the section might be misleading to some readers. The relevant finding for this peptide population is that the described proteogenomics approach shows comparable fraction of non-coding peptides from both rescoring and non-rescoring, thus indicating a robustly controlled FDR which is further substantiated by the HLA motif analysis. This is good stuff. :) Thus an appropriate title could be "Rescoring of melanoma immunopeptides shows robust FDR control in proteogenomics".

2d) MaxQuant vs other search engines

Rescoring improvement was shown only for MaxQuant as mentioned in the discussion. Thus, it is unclear whether the shown improvement generalizes to other search engines or is just a reflection of lower performance of MaxQuant in this context in the first place. If you have prior evidence of generalized improvement for all search engines from other Prosit models, it is fine to indicate this and that you expect the improvement this to hold for the current model as well. However, the statements in the abstract and discussion should be rephrased to inform the reader of this aspect to not misleadingly imply generalizability, e.g. "By applying our fragment ion intensity prediction model to MaxQuant results, we demonstrate up to 3-fold improvement. Based on previous studies [references], we expect improvement when applied to other search engines as well".

Minor revision points

3) "TimsTOF" should not be capitalized in the title.

4) Number of MS/MS spectra

Page 2 - 300,00 non-tryptic peptides were measured, but only 93,227 non-tryptic MS/MS spectra are reported (unless I misunderstand something here). It is surprising that only 1/3 or fewer of the synthesized peptides resulted in satisfactory MS/MS spectra. Is this because of very stringent quality requirements of the spectra? If so, please describe this briefly in the methods. Also, it would be important to know how many unique sequences are present in the 93,227 spectra.

5) Database search parameters

A more detailed description of relevant search parameters is necessary, even if the defaults were used. Default values can sometimes be surprisingly difficult to reproduce, and it makes it difficult to tell when a parameter might need adjusting or if it was just not reported by mistake.

Related to the search parameters, how many top PSMs per spectrum are reported by the search and used in the rescoring? What was the rationale or evidence in choosing the number? This would be important to know as more top PSMs are typically used in immunopeptidomics than in proteomics.

6) Oktoberfest

What is Oktoberfest? It is presented like an established tool, in which case it needs a reference, but I didn't find any references for it in the manuscript. If this is the first description of it, then it should be described more thoroughly than been done in the methods section.

7) Mirror plots

All the mirror plots are dominated by thick black lines that make it difficult to see the b-ion. It would be helpful if the unannotated ions were light grey, and the b-ions a darker hue so they stand out more clearly.

8) Ligand predictions

Why was binding affinity score used for ligand prediction instead of eluted ligand %rank? Eluted ligand is more relevant to ligand presentation, and %rank is more interpretable across multiple alleles than an affinity or score. This should also be explicitly called "% rank" in the figure labels, not "score".

9) Page 10 – "Thus, collision energy calibration is needed for accurate fragment ion intensity predictions."

This is the only time CE calibration is mentioned in the main text, and this whole paragraph seems out of place. We already know that collision energy has a profound impact on the information content of MS/MS spectra. If the purpose is to illustrate one reason why timsTOF and Orbitrap fragmentation spectra are different, then this paragraph would be better placed in the introduction section.

10) Rescoring features

The features added to the standard Percolator input should be briefly described. E.g. which and how many intensities were used in calculating similarity, what other additional values were calculated, etc?

11) Figure S3

Why are there no C or P amino acids at the C-termini in the test set?

12) Figure 2a

It would be helpful to add numbers above the bars. Due to the scale we cannot approximate the numbers in the validation and test sets from the bars alone.

13) Figure 3

No loss of identifications is visible. Please provide the number or explain why no loss has been observed in contrast to previous publications (Wilhelm et al., NatCom 2021, Figure S12b).

13a) Disconnect to previously reported increases

Prosit 2020 shows in Figure 3a an increase of 2.5 fold for CID and 1.4 for HCD, respectively. Wilhelm et al., NatCom 2021 reports on melanoma samples "...on average 3-fold more HLA class I peptides per patient ... and on average 2.4-fold for HLA class II peptides". Are there systematic differences in the

evaluation that can explain the lower increase, in particular for HCD?

13b) Confounding factors

The reported improvements depicted in Figure 3 need to differentiate relevant confounding factors. The slight improvement in performance difference between Orbitrap and TimsTOF might be due to the different charge state compositions of the results from the instruments. Thus, improvements should be reported separated by charge in line with the description of the data set (Figure 2a)

13c) PSM vs. Unique peptides

Figure 3 shows "Unique ligands" according to the y-axis label yet the figure legends talks about boosting "peptide-spectrum matches". Since peptides might be redundantly fragmented or occur in different charge states, it is relevant for the user to know how the performance differs for "unique peptides" and "PSMs". Thus, both plots should be provided.

13d) Negative control

Figure 3 is missing a negative control. Using the CID Prosit 2020 model on the timsTOF data and TOF Prosit 2023 on the Orbitrap data as a Supplementary figure is necessary to provide relevant support for the benefit depicted.

14) Figure 4

14a) The captions in figure 4 are rather confusing. It is not clear that the mono-allelic cell line data is only being used to make the reference motif. It reads like all the data, including the gained/shared/lost peptides, comes from mono-allelic cell lines instead of from A-375 cells. Please state where the gained/shared/lost peptides are coming from, and also clarify this in the main text. This also applies to Figure S4.

14b) If possible, it would be helpful to indicate the fold-changes above the paired bars in figure 4a. This can be done pretty elegantly with a square bracket connecting the two bars and a number above it, or possibly with a secondary y-axis (though that might be cluttered).

15) Model selection criteria

What was the reasoning to use the HCD Prosit 2020 model as the starting point for fine-tuning as opposed to the CID Prosit 2020 model or the subsequently released non-tryptic Prosit model?

Similarly, why was the HCD Prosit 2020 used for CE calibration? How did you assess its performance for this purpose (CE calibration of data from a timsTOF) compared to the other available models?

16) Ion mobility prediction

Is there any ion mobility prediction used in the rescoring? I would guess not as this is a fine-tuning of HCD Prosit 2020, and this the IMS would have to be a separate predictor, but given that ion mobility is orthogonal to retention time it would be quite interesting to see this.

18) nuORF source proteins

It is stated that there is a 2-fold increase in identified nuORF source proteins, but it is unclear which sample this was observed in. Does this refer to the 1-million cell sample? 40-million cell sample? All results combined?

19) Page 2 - "post-processing results from an unfiltered database search using machine learning algorithms"

This is nitpicky, but rescoring algorithms do not have to involve machine learning. PeptideProphet and iProphet are well-known tools which (to the best of my knowledge) use statistical approaches.

20) Page 2 - "Specifically, the use of Prosit led to a more than two-fold increase in the identification of HLA ligands"

"For example" might be better wording here. "Specifically" does not really follow from the previous sentence and implies that Prosit is the only thing out there for this.

21) Page 2 – “integration of fragment ion intensity predictions into the database searching process significantly improved the identification rate of HLA peptides”
Rescoring is not part of the database search process. It is a post-processing step. This would be better phrased as “search result rescoring process”

22) Page 6 – “We hypothesized that integrating fragment ion intensity predictions into the database searching process would improve the identification rate of HLA peptides”
See above.

23) Abstract – “Over 300,000 synthesized non-tryptic peptides from the ProteomeTools project were analyzed on a timsTOF-Pro to generate a dataset that was used to fine-tune an existing Prosit model.”
Please include number of MS2 spectra and unique peptide sequences used in training. These numbers are more relevant than the number of synthesized peptides.

Reviewer #2:

Adams et al. Fragment ion intensity prediction improves the identification rate of non-tryptic peptides in TimsTOF

Revision summary

Adams *et al.* adapted the existing Prosit model to predict peptide MS/MS fragmentation data, using timsTOF data and implemented it in Oktoberfest to rescore the MaxQuant PSMs. They prove that the (tims)TOF (CID) Prosit model outperforms the HCD-(Orbitrap)-Prosit model for predicting timsTOF MS/MS peptide spectra. Then, they show the benefits of this tool on previously published HLA class I and class II ligandomics (immunopeptidomics) data, increasing by up to 3-fold more unique peptides, depending on the dataset. In addition, the authors show that rescoring enhances the identification of non-canonical immunopeptides from nuORFs.

The authors clearly show the performance and benefits of their approach and also mention some of the limitations. The main innovation is adapting an MS/MS prediction model for tryptic and non-tryptic peptides based on data acquired from synthetic peptides, which can be considered a “ground truth.” Given the importance of MS-based immunopeptidomics for discovering therapy targets and the increasing usage of timsTOF instruments, this work is of high value to the community. However, some aspects require further evaluation, such as the effect of peptide size and type on the prediction and rescoring performance. I am confident that the authors will be able to address these issues. Thus, I recommend the publication of this manuscript after major revisions. Please find my comments and suggestions below.

Comments to the authors

Abstract

1. The following phrase seems incomplete. “Because the generation of immunopeptides from their parent proteins does not adhere to clear-cut rules, rather than being able to use known digestion patterns, every possible protein subsequence within HLA class-specific length restrictions needs to be considered” - into the database used for peptide identification?
2. It would be more informative to specify the number of acquired peptide spectra used for the model instead of or in addition to the total synthetic peptides. Please include it here.

Introduction

3. Paragraph 2. The value of including mutations, pathogen genomes, and nuORFs in the database for immunopeptide identification needs to be clarified here. Similarly, the importance of immunopeptides for immunotherapy is mentioned in the abstract, introduction, and discussion, but it is not explained why or how are they used. Although these aspects would be clear for experts in immunopeptidomics, they may not be for other readers. Thus, please briefly explain these two points.

Results general remarks

4. The authors named the new model “TOF Prosit 2023,” but I recommend renaming it “timsTOF Prosit 2023” to represent better the data used for training. Indeed, the performance of the model on data from other TOF instruments was not tested. Moreover, it has been observed in other instruments that the ion mobility step may influence peptide fragmentation (ref. 30), and it is possible that it also happens in the timsTOF, as correctly mentioned by the authors in the Discussion paragraph 3.
5. Peptide-HLA immunoprecipitated samples may contain peptides that are not necessarily HLA ligands. For instance, the antibody and HLA proteins may degrade into smaller peptides or other peptides may be co-enriched due to unspecific binding. Thus, I recommend adapting the labels of the figures (Fig. 3, Fig. 4a) and the text indicating the total number of peptides to “peptides” and using the terms “HLA-I ligands” and “HLA-II ligands” only for peptides that have been predicted to bind the correspondent HLA ligands. Although the prediction algorithms are imperfect, this provides a fairer comparison across the literature.
6. Tryptic peptides were also included in the training and test datasets. Although the performance on the test dataset was not evaluated in function of peptide types, the results indicate that the model could be used for both tryptic and non-tryptic peptides. Thus, it would be valuable to show if the identification of this type of peptide is also improved by rescoring with the timsTOF Prosit model. I suggest a dataset with at least two different

proteases to support the discussion (“... our model holds promise for numerous other biological and biomedical applications. One such area is deep proteome sequencing, where multiple proteases are used to enhance proteomic coverage”). It has been shown that immunopeptidomics MS/MS models may not translate so well to non-tryptic proteases and vice versa (Ref. 10, Declercq *et al.* 2021). Alternatively, a mixed proteome dataset could be used for validation since rescoring-rescued peptides should be associated with the corresponding species.

Results section 1

0. The authors analyzed an impressive number of 300,000 synthetic non-tryptic peptides, but only 93,227 were identified. (a) Can the authors explain why the identification rate was so low on the timsTOF instrument (31%), whereas the previous report (ref. 11) mentioned a synthesis/detection success ratio of 88%? (b) To better understand the nature of the peptides actually used for the model, a descriptive analysis should be included, at least in the supplementary material. For instance, the percentage of HLA class I binders, HLA class II binders, and AspN/LysN peptides; length distribution; and HLA allele coverage, e.g., against the twelve HLA supertype representatives from NetMHCpan 4.1 (<https://services.healthtech.dtu.dk/services/NetMHCpan-4.1/>).
1. Paragraph 2. Please mention here that the tryptic MS/MS spectra from ref. 2 were acquired using synthetic peptides. It was not clear to me until I looked for the reference.
2. Fig. 1. The spectra showed many high-intensity peaks not assigned to the peptide GVDAANSAAQQY, particularly on the Orbitrap side. This indicates that it could be a chimeric spectrum resulting from peptide coelution and co-isolation. Can the authors provide a cleaner example?
3. S1. Is the collision energy-Orbitrap in absolute values or normalized? Please specify it and add the units for both axes.

Results section 2

4. Saying “277,781 MS/MS spectra obtained in this study” is misleading since the tryptic peptides were not acquired in this study. I think “compiled” would be a better word than “obtained.”
0. (a) Which metric is used to conclude that “The TOF Prosit 2023 model demonstrates consistent performance across different precursor charges, peptide lengths, and collision energies (Supplementary Fig. S2a-c)”? The SA distribution differs slightly between charges (S2a). In addition, it decreases and becomes more spread in function of the peptide length. (b) Can the authors please elaborate on the reason for this behavior? (c) This should also be corrected in the footnote of Figure S2.
5. (a) Can the authors specify the proportion of peptides used for each dataset (training/validation/test)? This could be included in the text or as part of Fig 2. In addition, it seems that the proportion of singly charged peptides in the validation and test datasets

is lower than in the training (about 40%?). (b) Is this just a visual effect, or was the proportion of singly charged so low? Could this affect the model? It is essential to ensure the performance of the model for singly charged peptides, considering that they may represent an important part of the HLA-I ligandome in some cases (Refs. 14, 15, 23).

Results section 3

6. The increase in identifications is considerably smaller for peptides in the HLA-II dataset for both the Orbitrap and timsTOF. This is a limitation that is worth more attention. (a) Can the authors explain the possible reasons for the lower performance of Class II peptides? (b) Could it be derived from the peptide length-dependent effect observed in Fig. S2? (see point 12).

Results section 4

7. The experiment with different sample amounts performed by Phulphagar et al. [30] was done by injecting 1 to 40 million A-375 cell equivalents from a larger sample. To avoid confusing this with a true initial sample amount, the term “cells” should be changed to “cell equivalents” in this section.
8. The authors mention: “The benefit of rescoring is expected to be even higher for low input samples. To validate this assumption, we performed a reanalysis on this timsTOF. To validate this assumption, we performed a reanalysis on this timsTOF SCP dataset using the TOF Prosit 2023 model.” I would have assumed the same. However, the increase was lower for 1 million (1.3-fold) than 40 million cell equivalents (1.9). Could the authors comment on the possible reasons for this?

Discussion

9. Although the dataset used for the model was very large, it is not genuinely comprehensive (“including all or nearly all elements or aspects of something”). I would recommend using the word “extensive” instead of “comprehensive” in the phrase “[...] we established a comprehensive dataset [...].”
10. “Similarly to what has been observed for retention time alignment 31, we expect a benefit from collision energy alignment to account for the run-to-run fluctuations.” Is this something that was addressed in this manuscript or something that the authors plan to implement in the future? This was not clear to me; please clarify it.

Methods

11. Data acquisition / Synthetic non-tryptic peptides data acquisition / paragraph 2. (a) There seems to be an error in the description of the TIMS range used for the acquisition of non-tryptic peptide data. I downloaded some of the files and the range is 0.7 to 1.65 Vs/cm² and not 0.7 to 1.645 as written in the methodology. Please correct the

text. (b) In addition, the authors should mention that no isolation polygon was used to select precursors for MS/MS. Otherwise, one may assume that the standard polygon for proteomics was used, which would result in the exclusion of singly charged peptides. (c) Besides, some of the larger singly charged peptides have an inversed ion mobility above 1.65 Vs/cm² (See ref. 15). Thus, a TIMS range of 0.6 to 1.8 Vs/cm² could have resulted in more identifications. The authors should comment on this as a limitation in the Discussion.

12. a) Were the cysteine-containing peptides synthesized with carbamidomethylated cysteine, or were they treated to enforce this modification? Otherwise, why was carbamidomethylated cysteine used as a fixed modification?
- b) Contrary to proteomics workflows, immunopeptidomics preparation typically does not include the reduction and alkylation steps. Thus, using carbamidomethylated cysteine may negatively affect the detection of cysteine-containing HLA peptides (<https://doi.org/10.1038/s42003-021-02364-y>). The authors should evaluate this.

Other comments

13. Please add page and line numbers when resubmitting. It helps reviewers make comments on specific lines.
14. Define non-standard abbreviations the first time they are used. For instance, "SPD".
15. Some preprints are cited across the manuscript, which I see as a positive point. However, the references should specify their preprint status. For instance, this is correctly mentioned in references 12 and 39 but not in references 14 and 15. In addition, MSBooster was published during the revision time; please replace the preprint with the final reference <https://doi.org/10.1038/s41467-023-40129-9>.
16. I downloaded some of the files from the dataset PXD043844. When I unzip them, the raw file is borrowed in a series of folders. This may create problems when others try to repurpose the data. Please correct this if it is still possible. (e.g., HLA1_1_96_p2-A4_S2-A4_1_6929.d\Users\adams\Projects\non-tryptic-raw\HLA1_1_96_p2-A4_S2-A4_1_6929.d).

Reviewer #3:

Remarks to the Author:

This work's primary contribution is the generation of a large number of synthetic peptide spectra using the TimsTOF-Pro instrument. The data set could be used to fine-tune the Prosit model for fragment ion intensity prediction. The authors demonstrated that PSM rescoring using the fine-tuned Prosit could greatly improve the identification of immunopeptides. The data set of peptide spectra generated in this study must be an invaluable public resource and would be used in numerous studies. However, the methods presented by the authors are not novel; fragment ion prediction-based rescoring has been utilized in numerous studies. This study appears more appropriate for a data resource paper than a research article. To demonstrate utility, additional results should be provided.

Comments:

1. Figure 3 depicts the identification improvement resulting from rescoring based on fragment ion prediction. How did the authors identify the initial peptides for the comparison? In calculating FDR for the initial peptide identification, did the authors employ Andromeda scores without any rescoring? PSM rescoring (using Percolator, for example) has become widespread. The fragment ion prediction is not necessary for rescoring, which takes into account multiple other factors. Therefore, the improvement in peptide identification should be demonstrated by comparing rescoring results without and with fragment ion prediction. This comparison will clearly show the impact of fragment ion prediction. Figure 3 should also include the results of timsTOF - CID Prosit 2020, which will demonstrate the usefulness of the fine-tuned model using a synthetic timsTOF data set.

Response to the Review

Reviewer 1

The authors describe a fine-tuned version of the Prosit model for MS2 fragment ion intensity and precursor retention time prediction. The model is an iteration of the HCD Prosit 2020 model, fine-tuned on newly acquired and publicly available timsTOF MS2 fragmentation spectra of non-tryptic peptides. The model is tested on a publicly available benchmarking dataset and on newly acquired samples. The authors adequately demonstrate that timsTOF fragmentation spectra are considerably different from Orbitrap spectra, and that using the model as part of rescoring MaxQuant search results is demonstrated to increase identification rates.

I found the manuscript and methods easy to read and understand for the most part. I was convinced that the newly trained model is a valid and important contribution to the immunopeptidomics community. However, several points need to be addressed before the manuscript should be considered for publication. Beyond minor revisions to clarify findings, text and figures, there are two major points. First, the central theme of the manuscript is that it is fragment intensity prediction that is driving the performance of the model-based rescoring pipeline. However, the specific contribution of fragment ion intensity to the improvement after rescoring is never assessed independently of all other features contributing to rescoring. Second, the results tend to be overinterpreted and conclusions should be toned down to match the presented findings.

We thank the reviewer for the kind words and constructive comments. We have carefully reconsidered and reworked sections in the entire manuscript to clarify findings, adjust (over-)interpretations, and show the contribution of the fragment ion intensity to the identification rate improvement. Below are our replies to the specific comments.

Major revision points

1) Rescoring in general vs. fragment intensity prediction

The manuscript proposes an improvement in identification rate due to fragment ion intensity prediction. The improvement in identification rate is clearly shown in Figure 3 and 4, yet it is not shown that this is caused by fragment intensity prediction. What is shown is that the rescoring improves the identification rate, and the rescoring includes fragment intensity prediction, retention time prediction, and all the other features which are passed as inputs to Percolator. Thus, the contribution of fragment intensity prediction has not been assessed. The contribution of each factor (beyond the standard Percolator input) compared to a suitable reference (in this case MaxQuant and MaxQuant + the same rescoring pipeline presently used without any additional predictions) as well as the contribution of combinations of factors needs to be investigated before attributing factor-specific claims. Thus, Figure 3 needs to be extended to visualize the effect of the individual one- and two-factor models. Otherwise, the

wording throughout the manuscript needs to be adapted regarding this claim, e.g. “Rescoring improves identification rate...” instead of “Fragment ion intensity prediction improves...”.

This is a very valid comment. In the original submission we compared MaxQuant + Prosit + Percolator against MaxQuant + Percolator. However, this was unclear in the original manuscript and is adjusted for this version:

We compared results from MaxQuant + Percolator vs MaxQuant + Prosit + Percolator, respectively the feature sets can be found in Supplementary Table S1 and S2.

To investigate the impact of fragment ion intensity prediction, we performed MaxQuant + Prosit + Percolator again, however only with fragment ion intensity prediction based features, thus removing the retention time prediction based features. This was compared against MaxQuant + Percolator. We did this analysis with all the different models mentioned in this study: timsTOF Prosit 2023, HCD Prosit 2020, and CID Prosit 2020 model. We observed a consistently higher gain after rescoring if the timsTOF Prosit 2023 model was used to generate the spectral features. A paragraph was added to the results and two new panels were added to Figure 3:

To evaluate the effect of the fragment ion intensity prediction model, PSM rescoring was performed on the timsTOF data with the different models. To isolate the benefit of fragment ion intensity information during rescoring, RT prediction-based features were excluded during this analysis (Supplementary Table S1). We observed that PSM rescoring with the timsTOF 2023 model consistently resulted in a higher gain compared to PSM rescoring with the other two models (Fig. 3e-f). As expected the timsTOF Prosit 2023 model results in a higher gain of identifications compared to the HCD Prosit 2020 and the CID Prosit 2020 model. A possible explanation as to why PSM rescoring with the HCD Prosit 2020 model consistently results in a higher increase compared to PSM rescoring with the CID Prosit 2020 model, is that both HCD and timsTOF have a beam-type fragmentation and are thus more similar compared to CID.

Figure 3. Gained, shared, and lost peptide identifications for different sample types to compare PSM rescoring on Orbitrap data with PSM rescoring on timsTOF data. In general PSM rescoring was able to boost the confidence in peptide identifications, retaining true PSMs, gaining new PSMs, and losing only a small number of previously incorrect PSMs. **a** On average PSM rescoring of HLA-I Orbitrap data with the CID Prosit 2020 model resulted in a 2.2-fold increase (with 27,023 shared, 28,890 gained, and 486 lost unique peptides). **b** On average PSM rescoring of HLA-I timsTOF data with the timsTOF Prosit 2023 model resulted in a 2.7-fold increase (with 41,822 shared, 62,962 gained, and 521 lost unique peptides). **c** On average PSM rescoring of HLA-II Orbitrap data with the HCD Prosit 2020 model resulted in a 1.4-fold increase (with 30,081 shared, 12,187 gained, and 537 lost unique peptides). **d** On average PSM rescoring of HLA-II timsTOF data with the timsTOF Prosit 2023 model resulted in a 1.8-fold increase (with 47,201 shared, 26,852 gained, and 787 lost unique peptides). **e** To evaluate the effect of the fragment ion intensity prediction model on PSM rescoring, the RT prediction-based features were excluded. On average PSM rescoring of HLA-I timsTOF data with the timsTOF Prosit 2023 model resulted in a 2.8-fold increase (with 39,606 shared, 59,458 gained, and 549 lost unique peptides). On average PSM rescoring of HLA-I timsTOF data with the HCD Prosit 2020 model resulted in a 2.5-fold increase (with 39,296 shared, 52,520 gained, and 859 lost unique peptides). On average PSM rescoring of HLA-I timsTOF data with the CID Prosit 2020 model resulted in a 2.2-fold increase (with 38,431 shared, 43,748 gained, and 1,724 lost unique peptides). **f** To evaluate the effect of the fragment ion intensity prediction model on PSM rescoring, the RT prediction-based features were excluded. On average PSM rescoring of HLA-II timsTOF data with the timsTOF Prosit 2023 model resulted in a 1.5-fold increase (with 46,439 shared, 25,443 gained, and 918 lost unique peptides). On average PSM rescoring of HLA-II timsTOF data with the HCD Prosit 2020 model resulted in a 1.5-fold increase (with 45,393 shared, 24,250 gained, and 964 lost unique peptides). On average PSM rescoring of HLA-II timsTOF data with the CID Prosit 2020 model resulted in a 1.5-fold increase (with 46,088 shared, 21,899 gained, and 1,269 lost unique peptides). RCC = renal cell carcinoma; HNSCC = head and neck squamous-cell carcinoma; PBMC = peripheral blood mononuclear cell; CLL = chronic lymphocytic leukemia.

2) Result overinterpretation

Interpretation of results is exaggerated and should be down-toned to match the described findings. The following points should be addressed.

Thank you for bringing these points to our attention. We put great effort into adjusting the manuscript where necessary to avoid exaggerated interpretations of the results.

2a) Accurate prediction

The second paragraph in the result section is headlined with “model allows accurate prediction”. It should be rather rephrased as “model improves prediction accuracy” unless it can be shown that the accuracy achieved is in-line with the variation observed for technical replicates.

Thank you for this suggestion, we changed the subtitle accordingly:

Optimized Prosit model **improves prediction accuracy** of tryptic and non-tryptic peptide timsTOF MS/MS spectra

2b) TimsTOF vs. Orbitrap

The third paragraph states that “rescoring boosts identification on timsTOF compared to Orbitrap”. The findings, however, show that “rescoring boosts identification on timsTOFs as well as Orbitrap”. The difference is 2.8 vs. 2.5 and 1.7 vs. 1.4 fold for timsTOF vs. Orbitrap rescoring. It is not shown that this average improvement by 0.3 is significant. These results show the power of rescoring and that this power can be utilized for both MS platforms, but it isn’t clear that there is a substantial difference between the platforms. A helpful comparison here could be showing how the using the timsTOF model on timsTOF data improves the identification rate compared to using the orbitrap model on timsTOF data. This would very directly show the improvement when using an instrument-specific model.

Thank you for this valuable suggestion. We agree that it would be a useful comparison to look at the improvement in identification rate between rescoring using the timsTOF Prosit 2023 model compared to rescoring using the Orbitrap models. To evaluate the effect of the fragment ion intensity predictions, we removed the RT prediction-based features and performed rescoring using the timsTOF model and the two Orbitrap models mentioned in this study (see updated Figure 3). The subtitle was changed accordingly:

PSM rescoring boosts immunopeptide identification on timsTOF ~~compared to Orbitrap~~

The manuscript was also adjusted accordingly:

To evaluate the effect of the fragment ion intensity prediction model, PSM rescoring was performed on the timsTOF data with the different models. In contrast to the PSM rescoring approach used above, the RT prediction-based features were not included in this analysis. We observed that PSM rescoring with the timsTOF 2023 model consistently resulted in a higher gain compared to rescoring with the other two models (Fig. 3e-f).

2c) Novel neo-epitopes

The fourth results section claims that rescoring “reveals novel neo-epitopes”. However, the description describes a proteogenomics search for the identification of peptides potentially derived from a non-coding region. These should be referred to as non-canonical peptides, not neoepitopes. Whole-exome sequencing is necessary to define somatic mutations (neoantigens) which give rise to neoepitopes, and this is absent. While the term neoepitopes is misused in various publications, its definition in the context of cancer is clear. See, for instance, Chong et al., NatCom 2020 (PMID 32157095).

The term “neo-epitopes” was removed from the manuscript.

Furthermore, the attribute “novel” is not substantiated. There is neither a comparison with previous publications (if the identified peptide candidates have been described elsewhere) nor is there an experimental verification that the peptides identified with statistical uncertainty are

supported by mRNA evidence or synthetic spike-in experiments standard in the field (e.g. Skipper et al., Int J Cancer 1999, PMID 10417764).

The adjective “novel” was removed where it was not substantiated.

From the second paragraph of this section: “Consequently, the benefit of rescoring is expected to be even higher for low input samples. To validate this assumption, we performed a reanalysis on this timsTOF SCP dataset using the TOF Prosit 2023 model.” – The results of the titration series, which directly follow this passage, show the opposite of this assumption. It is shown that the largest fold-increase is achieved for the highest-input sample, and there appears to be a trend of lessening fold-increase with lessening sample input. Further, the results described in this section indicate a lower fold-increase than was observed when rescoring the benchmarking study, counter to what we are led to expect. Of course, it is not a problem to have data that doesn’t support a specific hypothesis, but if it actually is a hypothesis you intended to test then please add some interpretation as to why the results are counter to expectations. If it isn’t a hypothesis, but just an introductory sentence to the analysis, it might be better to phrase it as an evaluation of the rescoring on low-input data, rather than validation of an assumption of how the rescoring will perform.

We evaluated why our results did not support this hypothesis and acknowledge that the initial hypothesis was incorrect. It stated that low sample inputs frequently suffer from missing peaks, as fragments with low intensities fail to surpass the noise level and that this leads to low database search engine scores and low identification rates. However, as we mention in the introduction, MS/MS spectra from timsTOF instruments exhibit more reproducibility at low abundances compared to MS/MS spectra from Orbitrap instruments. Consequently, we do not necessarily expect low database search engine scores and low identification rates.

This is supported by the observation that MaxQuant + Percolator already results in the annotation of more than half the MS/MS spectra in the 1 million cell equivalent sample. When looking at the distributions of the targets and decoys, it seems there are not many targets left to identify:

Additionally, to validate whether the spectra were stable in low sample inputs, we assessed the spectral angle for all cell equivalents and observed a consistent median spectral angle around 0.85.

The initial hypothesis was removed from the manuscript and the following statement and supplementary figure were added:

Interestingly, we observed a consistent median spectral angle around 0.85 across all cell equivalents, supporting the finding that MS/MS spectra from timsTOF instruments are reproducible at low abundances¹⁷ (Supplementary Fig. S8b).

Lastly, since rescoring was shown to increase the total number of identifications, it is expected that there will be an absolute higher number of non-canonical peptides identified as well. This effect could be reported in any peptide population investigated and is not specific to non-canonical peptides, so the title of the section might be misleading to some readers. The relevant finding for this peptide population is that the described proteogenomics approach shows comparable fraction of non-coding peptides from both rescoring and non-rescoring, thus indicating a robustly controlled FDR which is further substantiated by the HLA motif analysis. This is good stuff. :) Thus an appropriate title could be “Rescoring of melanoma immunopeptides shows robust FDR control in proteogenomics”.

To address this comment, we updated the text to highlight this aspect more:

Notably, **although** we did not observe significant changes in the ratio of nuORFs after PSM rescoring, **indicating a robust FDR control in proteogenomics**.

The title was updated to:

PSM rescoring **boosts the identification rate of relevant immunopeptides in low-input samples**

2d) MaxQuant vs other search engines

Rescoring improvement was shown only for MaxQuant as mentioned in the discussion. Thus, it is unclear whether the shown improvement generalizes to other search engines or is just a reflection of lower performance of MaxQuant in this context in the first place. If you have prior

evidence of generalized improvement for all search engines from other ProSist models, it is fine to indicate this and that you expect the improvement this to hold for the current model as well. However, the statements in the abstract and discussion should be rephrased to inform the reader of this aspect to not misleadingly imply generalizability, e.g. "By applying our fragment ion intensity prediction model to MaxQuant results, we demonstrate up to 3-fold improvement. Based on previous studies [references], we expect improvement when applied to other search engines as well".

Oktoberfest supports several search engines (MSAmanda, Mascot, MSFragger) and have shown similar improvement across search engines, using the Orbitrap models. The discussion was updated to emphasize that MaxQuant was used throughout this manuscript:

By employing this model for PSM rescoring of MaxQuant results from immunopeptides measured on a timsTOF, we achieved a nearly 3-fold increase in the identification of HLA-I peptides. Based on previous studies ^{11,12,34}, we expect improvement when applied to other search engines as well.

Minor revision points

3) "TimsTOF" should not be capitalized in the title.

We adjusted the title.

4) Number of MS/MS spectra

Page 2 - 300,00 non-tryptic peptides were measured, but only 93,227 non-tryptic MS/MS spectra are reported (unless I misunderstand something here). It is surprising that only 1/3 or fewer of the synthesized peptides resulted in satisfactory MS/MS spectra. Is this because of very stringent quality requirements of the spectra? If so, please describe this briefly in the methods. Also, it would be important to know how many unique sequences are present in the 93,227 spectra.

This is a pertinent comment, which we are happy to clarify. We believe that this is a very valuable dataset that we expect will be reused by others. Within the ProteomeTools project, 302,105 unique non-tryptic peptides were synthesized. These were all measured in pools on the timsTOF-Pro, resulting in 360,448 MS/MS spectra. After selecting for a PEP < 0.01, 260,151 (about 72%) of MS/MS spectra remained. After selecting for an Andromeda score ≥ 70, only 93,227 of MS/MS spectra remained. We briefly described our selection criteria in the methods section:

PSMs were filtered at a 0.01 posterior error probability (PEP). Only peptides expected in the pool, including full-length and N-terminally truncated peptides, were selected. All PSMs, even for the same peptide, with an Andromeda score ≥ 70 were included.

There were 73,325 unique peptide sequences present in the 93,227 MS/MS spectra acquired for the non-tryptic fraction of the training data. The number of unique peptide sequences of the data was added to the first result section:

The considerable dissimilarity in MS/MS spectra generated by timsTOF and Orbitrap instruments for the same peptide (Fig. 1, Supplementary Fig. S1) underscores the need to develop fragment ion intensity prediction models optimized for timsTOF data. To address this, we measured 302,105 unique synthesized non-tryptic peptides from the ProteomeTools project¹². Our measurements encompassed a range of collision energies from 20.81 eV to 69.77 eV, enabling us to investigate the impact of collision energy on fragment ion intensities. Consequently, we compiled a dataset consisting of 93,227 non-tryptic MS/MS spectra, complemented by 184,552 previously published tryptic MS/MS spectra from 138,227 unique synthetic tryptic peptides²². This extensive dataset, comprising a total of 277,779 MS/MS spectra and 216,934 unique peptides, serves as a unique training dataset for the development of machine learning tools tailored to timsTOF instruments.

5) Database search parameters

A more detailed description of relevant search parameters is necessary, even if the defaults were used. Default values can sometimes be surprisingly difficult to reproduce, and it makes it difficult to tell when a parameter might need adjusting or if it was just not reported by mistake.

Related to the search parameters, how many top PSMs per spectrum are reported by the search and used in the rescoring? What was the rationale or evidence in choosing the number? This would be important to know as more top PSMs are typically used in immunopeptidomics than in proteomics.

To ensure reproducibility the mqpar files containing all search parameters were included in the publicly available data together with the MaxQuant output folder. As peptide pools were explicitly constructed to avoid overlapping near-isobaric peptides, there should not be an excessive rate of chimeric spectra. Hence, only the top PSM was used per spectrum. The methods section was updated to include more details on the used parameters:

Preparation of the training data

Default parameters were used, unless mentioned otherwise: carbamidomethylated cysteine was specified as a fixed modification and methionine oxidation as a variable modification. The minimal sequence length was set to 7 and the maximum sequence length was set to the maximum length of peptides in the pool. The precursor tolerance was set to 40 ppm and only the top PSM was used per spectrum.

Re-analysis of the comparison dataset

Carbamidomethylated cysteine was not included as a fixed modification, because cysteine was not carbamidomethylated during sample processing. Before PSM rescoring all PSMs containing free cysteine side chains were removed. The Orbitrap data was searched with a precursor tolerance of 20 ppm and the timsTOF data with a precursor tolerance of 40 ppm.

6) Oktoberfest

What is Oktoberfest? It is presented like an established tool, in which case it needs a reference, but I didn't find any references for it in the manuscript. If this is the first description of it, then it should be described more thoroughly than been done in the methods section.

During the reviewing process of this manuscript, the paper for Oktoberfest was accepted and published. The reference has been added in this revision:

26. Picciani, M. *et al.* Oktoberfest: Open-source spectral library generation and rescoring pipeline based on Prosit. *PROTEOMICS* 2300112 (2023).

7) Mirror plots

All the mirror plots are dominated by thick black lines that make it difficult to see the b-ion. It would be helpful if the unannotated ions were light grey, and the b-ions a darker hue so they stand out more clearly.

To improve visibility of the b-ions in the mirror plots, we adjusted the colors.

8) Ligand predictions

Why was binding affinity score used for ligand prediction instead of eluted ligand %rank? Eluted ligand is more relevant to ligand presentation, and %rank is more interpretable across multiple alleles than an affinity or score. This should also be explicitly called “% rank” in the figure labels, not “score”.

Thank you for bringing this to our attention. We checked this in the data and we actually did use the eluted ligand percentile rank. However, this should have been more clearly stated both in the results and in the methods. We updated the figure and text to specify that the percentile rank was used:

For each peptide we calculated the binding affinity to every HLA allele present in the cell line, using NetMHCpan version 4.1³³. For each peptide the best, i.e. lowest, **percentile rank value** was retained. A percentile rank cutoff of 2 was used for weak binders and 0.5 for strong binders³³.

9) Page 10 – “Thus, collision energy calibration is needed for accurate fragment ion intensity predictions.”

This is the only time CE calibration is mentioned in the main text, and this whole paragraph seems out of place. We already know that collision energy has a profound impact on the information content of MS/MS spectra. If the purpose is to illustrate one reason why timsTOF and Orbitrap fragmentation spectra are different, then this paragraph would be better placed in the introduction section.

The purpose is to highlight how important the CE calibration step is. To make this point clearer it was moved to the second section of the results:

It is important to note that the applied collision energy has a profound impact on the information content of the obtained MS/MS spectra²⁴. To optimize the transfer learning, the collision energies of the training, validation, and test set were calibrated according to the collision energies the HCD Prosit 2020 model would expect. To achieve this, a robust linear model was trained on the entire training set for each precursor charge state and for tryptic and non-tryptic peptides separately. The acquired linear models were applied on the training, validation, and test set. To ensure accurate fragment ion intensity predictions, collision energy calibration is needed.

In addition, a description on how CE calibration was performed for PSM rescoring was added to the methods:

Collision energies were calibrated by predicting the top 100 scoring PSMs for each charge state with a collision energy range between 5 and 45 eV. The SA was calculated between the observed spectra and the predicted spectra and the optimal collision energy was determined by selecting the collision energy corresponding to the top-scoring predicted spectra. A robust linear model was trained using RANSAC regression in scikit-learn version 1.2.2⁴⁹ to predict the difference between the reported collision energy and the optimal collision energy, based on the peptide mass. Subsequently, this model was used to determine the optimal collision energy.

10) Rescoring features

The features added to the standard Percolator input should be briefly described. E.g. which and how many intensities were used in calculating similarity, what other additional values were calculated, etc?

We agree that which features were used and how they are calculated should be clarified. We have added the reference to Oktoberfest to guide readers to this information. In addition, we

have included lists of used features for “MaxQuant + Rescoring” (Supplementary Table S1) and for “MaxQuant + Prosit + Rescoring” (Supplementary Table S2).

11) Figure S3

Why are there no C or P amino acids at the C-termini in the test set?

In the training set, 365 PSMs had C at the C-terminus and 240 had P at the C-terminus. In the validation set, 205 PSMs had C at the C-terminus and 51 had P at the C-terminus. In the test-set, however, there were no PSMs with either a C or a P at the C-terminus. To avoid data leakage the split was performed by selecting random pools. There were only 3 pools containing sequences with a C at the C-terminus and only 7 with a P at the C-terminus. Coincidentally, none of these pools were selected for the test set, resulting in no PSMs with either P or C at the C-terminus to be present in the test set.

12) Figure 2a

It would be helpful to add numbers above the bars. Due to the scale we cannot approximate the numbers in the validation and test sets from the bars alone.

Figure 2a was updated to include numbers above the bars:

13) Figure 3

No loss of identifications is visible. Please provide the number or explain why no loss has been observed in contrast to previous publications (Wilhelm et al., NatCom 2021, Figure S12b).

There is a loss of identifications. However, because the number of lost identifications is very low, this is not clearly visible in the figure. To address this we reported the total number of gained, shared, and lost unique peptides for each subplot in the figure description:

Figure 3. Gained, shared, and lost peptide identifications for different sample types to compare PSM rescoring on Orbitrap data with PSM rescoring on timsTOF data. In general PSM rescoring was able to boost the confidence in peptide identifications, retaining true PSMs, gaining new PSMs, and losing only a small number of previously incorrect PSMs. **a** On average PSM rescoring of HLA-I Orbitrap data with the CID Prosit 2020 model resulted in a 2.2-fold increase (with 27,023 shared, 28,890 gained, and 486 lost unique peptides). **b** On average PSM rescoring of HLA-I timsTOF data with the timsTOF Prosit 2023 model resulted in a 2.7-fold increase (with 41,822 shared, 62,962 gained, and 521 lost unique peptides). **c** On average PSM rescoring of HLA-II Orbitrap data with the HCD Prosit 2020 model resulted in a 1.4-fold increase (with 30,081 shared, 12,187 gained, and 537 lost unique peptides). **d** On average PSM rescoring of HLA-II timsTOF data with the timsTOF Prosit 2023 model resulted in a 1.8-fold increase (with 47,201 shared, 26,852 gained, and 787 lost unique peptides). **e** To evaluate the effect of the fragment ion intensity prediction model on PSM rescoring, the RT prediction-based features were excluded. On average PSM rescoring of HLA-I timsTOF data with the timsTOF Prosit 2023 model resulted in a 2.8-fold increase (with 39,606 shared, 59,458 gained, and 549 lost unique peptides). On average PSM rescoring of HLA-I timsTOF data with the HCD Prosit 2020 model resulted in a 2.5-fold increase (with 39,296 shared, 52,520 gained, and 859 lost unique peptides). On average PSM rescoring of HLA-I timsTOF data with the CID Prosit 2020 model resulted in a 2.2-fold increase (with 38,431 shared, 43,748 gained, and 1,724 lost unique peptides). **f** To evaluate the effect of the fragment ion intensity prediction model on PSM rescoring, the RT prediction-based features were excluded. On average PSM rescoring of HLA-II timsTOF data with the timsTOF Prosit 2023 model resulted in a 1.5-fold increase (with 46,439 shared, 25,443 gained, and 918 lost unique peptides). On average PSM rescoring of HLA-II timsTOF data with the HCD Prosit 2020 model resulted in a 1.5-fold increase (with 45,393 shared, 24,250 gained, and 964 lost unique peptides). On average PSM rescoring of HLA-II timsTOF data with the CID Prosit 2020 model resulted in a 1.5-fold increase (with 46,088 shared, 21,899 gained, and 1,269 lost unique peptides). RCC = renal cell carcinoma; HNSCC = head and neck squamous-cell carcinoma; PBMC = peripheral blood mononuclear cell; CLL = chronic lymphocytic leukemia.

13a) Disconnect to previously reported increases

Prosit 2020 shows in Figure 3a an increase of 2.5 fold for CID and 1.4 for HCD, respectively. Wilhelm et al., NatCom 2021 reports on melanoma samples "...on average 3-fold more HLA class I peptides per patient ... and on average 2.4-fold for HLA class II peptides". Are there systematic differences in the evaluation that can explain the lower increase, in particular for HCD?

There are no systematic differences in the evaluation, which was performed similarly as previously. A likely explanation on why the fold increase is lower compared to the reports by Wilhelm et al. on melanoma samples, is that gains after rescoring will vary for different datasets.

13b) Confounding factors

The reported improvements depicted in Figure 3 need to differentiate relevant confounding factors. The slight improvement in performance difference between Orbitrap and TimsTOF

might be due to the different charge state compositions of the results from the instruments. Thus, improvements should be reported separated by charge in line with the description of the data set (Figure 2a).

The reported improvements were evaluated for each charge state. Even though there are more identifications at charge state 1 for HLA-I peptides measured on timsTOF, the relative gains after PSM rescoring with Prosit predictions stays stable across charge states.

13c) PSM vs. Unique peptides

Figure 3 shows “Unique ligands” according to the y-axis label yet the figure legends talks about boosting “peptide-spectrum matches”. Since peptides might be redundantly fragmented or occur in different charge states, it is relevant for the user to know how the performance differs for “unique peptides” and “PSMs”. Thus, both plots should be provided.

Figure 3 with “PSMs” instead of “unique peptides” was added as a Supplementary figure:

Supplementary Figure S5. Gained, shared, and lost identified PSMs for different sample types to compare PSM rescoring on Orbitrap data with PSM rescoring on timsTOF data.

In general PSM rescoring was able to boost the number of PSMs, retaining true PSMs, gaining new PSMs, and losing only a small number of previously incorrect PSMs. **a** On average PSM rescoring of HLA-I Orbitrap data with the CID Prosit 2020 model resulted in a 1.9-fold increase (with 145,280 shared, 118,453 gained, and 3,466 lost PSMs). **b** On average PSM rescoring of HLA-I timsTOF data with the timsTOF Prosit 2023 model resulted in a 3.6-fold increase (with 89,098 shared, 163,852 gained, and 863 lost PSMs). **c** On average PSM rescoring of HLA-II Orbitrap data with the HCD Prosit 2020 model resulted in a 1.3-fold increase (with 165,160 shared, 44,796 gained, and 2,749 lost PSMs). **d** On average PSM rescoring of HLA-II timsTOF data with the timsTOF Prosit 2023 model resulted in a 2.3-fold increase (with 123,232 shared, 74,587 gained, and 1,328 lost PSMs). **e** To evaluate the effect of the fragment ion intensity prediction model on PSM rescoring, the RT prediction-based features were excluded. On average PSM rescoring of HLA-I timsTOF data with the timsTOF Prosit 2023 model resulted in a 3.4-fold increase (with 69,832 shared, 140,444 gained, and 5,437 lost PSMs). On average PSM rescoring of HLA-I timsTOF data with the HCD Prosit 2020 model resulted in a 2.9-fold increase (with 68,802 shared, 122,033 gained, and 6,467 lost PSMs). On average PSM rescoring of HLA-I timsTOF data with the CID Prosit 2020 model resulted in a 2.5-fold increase (with 66,745 shared, 101,782 gained, and 8,524 lost PSMs). **f** To evaluate the effect of the fragment ion intensity prediction model on PSM rescoring, the RT prediction-based features were excluded. On average PSM rescoring of HLA-II timsTOF data with the timsTOF Prosit 2023 model resulted in a 1.6-fold increase (with 91,281 shared, 66,874 gained, and 10,256 lost PSMs). On average PSM rescoring of HLA-II timsTOF data with the HCD Prosit 2020 model resulted in a 1.6-fold increase (with 91,184 shared, 63,897 gained, and 10,353 lost PSMs). On average PSM rescoring of HLA-II timsTOF data with the CID Prosit 2020 model resulted in a 1.5-fold increase (with 91,111 shared, 57,688 gained, and 10,426 lost PSMs). RCC = renal cell carcinoma; HNSCC = head and neck squamous-cell carcinoma; PBMC = peripheral blood mononuclear cell; CLL = chronic lymphocytic leukemia.

13d) Negative control

Figure 3 is missing a negative control. Using the CID Prosit 2020 model on the timsTOF data and TOF Prosit 2023 on the Orbitrap data as a Supplementary figure is necessary to provide relevant support for the benefit depicted.

Figure 3 was updated to show the effect of the different fragment ion intensity prediction models on the identification rate of the timsTOF data (see previously). In addition, we evaluated the effect of the different models on the identification rate of the Orbitrap data.

The evaluation of the effect of the different fragment ion intensity prediction models on PSM rescoring of the Orbitrap data achieved similar results, with the CID Prosit 2020 model consistently resulting in a higher gain of identifications in the HLA-I dataset and the HCD Prosit 2020 model consistently resulting in a higher gain of identifications in the HLA-II dataset (Supplementary Fig. S6).

Supplementary Figure S6. Gained, shared, and lost peptide identifications for PSM rescoring with different fragment ion intensity prediction models. In general PSM rescoring was able to boost the number of PSMs, retaining true PSMs, gaining new PSMs, and losing only a small number of previously incorrect PSMs. **a** To evaluate the effect of the fragment ion intensity prediction model on PSM rescoring, the RT prediction-based features were excluded. On average PSM rescoring of HLA-I Orbitrap data with the timsTOF Prosit 2023 model resulted in a 2.0-fold increase (with 26,994 shared, 27,056 gained, and 511 lost PSMs). On average PSM rescoring of HLA-I timsTOF data with the HCD Prosit 2020 model resulted in a 1.9-fold increase (with 26,775 shared, 24,137 gained, and 730 lost PSMs). On average PSM rescoring of HLA-I timsTOF data with the CID Prosit 2020 model resulted in a 2.3-fold increase (with 27,009 shared, 32,462 gained, and 496 lost PSMs). **b** To evaluate the effect of the fragment ion intensity prediction model on PSM rescoring, the RT prediction-based features were excluded. On average PSM rescoring of HLA-II timsTOF data with the timsTOF Prosit 2023 model resulted in a 1.4-fold increase (with 29,945 shared, 11,305 gained, and 569 lost PSMs). On average PSM rescoring of HLA-II timsTOF data with the HCD Prosit 2020 model resulted in a 1.4-fold increase (with 29,947 shared, 11,742 gained, and 567 lost PSMs). On average PSM rescoring of HLA-II timsTOF data with the CID Prosit 2020 model resulted in a 1.3-fold increase (with 29,605 shared, 9,162 gained, and 909 lost PSMs). RCC = renal cell carcinoma; HNSCC = head and neck squamous-cell carcinoma; PBMC = peripheral blood mononuclear cell; CLL = chronic lymphocytic leukemia.

14) Figure 4

14a) The captions in figure 4 are rather confusing. It is not clear that the mono-allelic cell line data is only being used to make the reference motif. It reads like all the data, including the gained/shared/lost peptides, comes from mono-allelic cell lines instead of from A-375 cells. Please state where the gained/shared/lost peptides are coming from, and also clarify this in the main text. This also applies to Figure S4.

Thank you for this suggestion. We have clarified the captions from Fig. 4b and Fig. S7:

Figure 4. PSM rescoring timsTOF SCP immunopeptidomics data. b Peptide motif plots of 1,406 unique peptides confidently identified to be present in the cell line expressing allele A*01:01³⁵, and peptide motif plots of 16,641 unique gained peptides, 13,208 unique shared peptides, and 447 unique lost peptides resulting from PSM rescoring of the 1 million to 40 million A-375 cell equivalents. Amino acids are colored according to their physicochemical properties (red acidic, blue basic, black hydrophobic, purple neutral, and green polar amino acids).

14b) If possible, it would be helpful to indicate the fold-changes above the paired bars in figure 4a. This can be done pretty elegantly with a square bracket connecting the two bars and a number above it, or possibly with a secondary y-axis (though that might be cluttered).

To improve the interpretability of the figure, we have added the fold-changes for each MaxQuant + Percolator and MaxQuant + Prosit + Percolator pair, in both Fig. 4a and Fig. S6a.

Figure 4. PSM rescoring timsTOF SCP immunopeptidomics data. a Mean and standard deviation showing the identification rates of 1 million to 40 million A-375 cells equivalents. Above each bar the fold change is shown between MaxQuant + Percolator and MaxQuant + Prosit + Percolator.

Supplementary figure S8. a Mean and standard deviation showing the number of unique HLA-I peptides identified from 1 million to 40 million A-375 cell equivalents. Above each bar the fold change is shown between MaxQuant + Percolator and MaxQuant + Prosit + Percolator.

15) Model selection criteria

What was the reasoning to use the HCD Prosit 2020 model as the starting point for fine-tuning as opposed to the CID Prosit 2020 model or the subsequently released non-tryptic Prosit model?

This is a very pertinent question. Both the HCD Prosit 2020 model and the CID Prosit 2020 model were trained on data from both tryptic and non-tryptic peptides. The reasoning for selecting the HCD Prosit 2020 model is that HCD is more similar to the collision induced dissociation used in a time-of-flight (TOF) instrument. This reasoning was also added in the manuscript:

The HCD Prosit 2020 model was selected because HCD is a non-resonant activation technique like the collision induced dissociation conducted in a TOF instrument ²³.

In addition, the CID Prosit 2020 model does not take into account the collision energy. In CID, even at different collision energies the fragment ion intensities will be very similar. In contrast, collision energy calibration is very important for the HCD Prosit 2020 model and was similarly expected to be important for a timsTOF fragment ion intensity prediction model, providing an additional reason to select HCD Prosit 2020, rather than CID Prosit 2020 as a starting point.

Similarly, why was the HCD Prosit 2020 used for CE calibration? How did you assess its performance for this purpose (CE calibration of data from a timsTOF) compared to the other available models?

This is a very relevant question, because CE calibration is a key element for getting accurate fragment ion intensity predictions. A paragraph was added to the results section:

It is important to note that the applied collision energy has a profound impact on the information content of the obtained MS/MS spectra ²⁴. To optimize the transfer learning, the collision energies of the training, validation, and test set were calibrated according to the collision energies the HCD Prosit 2020 model would expect. To achieve this, robust linear models were trained on the training set, stratified by precursor charge state and tryptic status, and then applied on the training, validation, and test sets (see Methods).

16) Ion mobility prediction

Is there any ion mobility prediction used in the rescoring? I would guess not as this is a fine-tuning of HCD Prosit 2020, and this the IMS would have to be a separate predictor, but given that ion mobility is orthogonal to retention time it would be quite interesting to see this.

As you have stated CCS/IM prediction-based features could be very relevant. However, we did not include this in the PSM rescoring yet. We have added a section on this in the discussion:

Another potential application of our timsTOF ground-truth dataset could be to develop a CCS prediction model for non-tryptic peptides. Analogous to fragment ion intensity and RT predictions, CCS predictions can be used as features during PSM rescoring ³⁸. Some CCS prediction models already exist ^{22,39,40}, including a recent model trained on immunopeptides ⁴⁰.

18) nuORF source proteins

It is stated that there is a 2-fold increase in identified nuORF source proteins, but it is unclear which sample this was observed in. Does this refer to the 1-million cell sample? 40-million cell sample? All results combined?

The 2-fold increase referred to all results combined. However, when we looked at the data separately, we found that this 2-fold increase is not representative for the different cell equivalents. We have adjusted the text accordingly:

The results showed an average increase in identified HLA-I ligands across different input sizes, ranging from 1.3-fold at 1 million cell equivalents to 1.9-fold at 40 million cell equivalents (Fig. 4a, Supplementary Fig. S5a).

19) Page 2 - "post-processing results from an unfiltered database search using machine learning algorithms"

This is nitpicky, but rescoring algorithms do not have to involve machine learning. PeptideProphet and iProphet are well-known tools which (to the best of my knowledge) use statistical approaches.

We have removed "machine learning" to not exclude post-processing tools like PeptideProphet.

20) Page 2 - "Specifically, the use of Prosit led to a more than two-fold increase in the identification of HLA ligands"

"For example" might be better wording here. "Specifically" does not really follow from the previous sentence and implies that Prosit is the only thing out there for this.

We have changed "Specifically" to "For example".

21) Page 2 - "integration of fragment ion intensity predictions into the database searching process significantly improved the identification rate of HLA peptides"

Rescoring is not part of the database search process. It is a post-processing step. This would be better phrased as "search result rescoring process"

We have changed the sentence accordingly:

We hypothesized that integrating fragment ion intensity predictions into PSM rescoring of search results would improve the identification rate of HLA peptides measured on a timsTOF, similar to what was previously observed for tryptic and non-tryptic peptides measured on other instruments¹¹.

22) Page 6 - "We hypothesized that integrating fragment ion intensity predictions into the database searching process would improve the identification rate of HLA peptides"

See above.

We have changed the sentence accordingly:

The integration of fragment ion intensity predictions into **PSM rescoring of search results** significantly improved the identification rate of HLA peptides measured on a timsTOF compared to an Orbitrap.

23) Abstract – “Over 300,000 synthesized non-tryptic peptides from the ProteomeTools project were analyzed on a timsTOF-Pro to generate a dataset that was used to fine-tune an existing Prosit model.”

Please include number of MS2 spectra and unique peptide sequences used in training. These numbers are more relevant than the number of synthesized peptides.

Thank you for this suggestion. We intend to highlight the usefulness of this dataset and aim to make it reach people working on ML-based approaches for timsTOF. We want the training data to be represented well and it is a very valid point that the number of confidently identified MS/MS spectra and the number of unique peptide sequences is more relevant than the number of synthesized peptides. We adjusted the text accordingly:

302,105 unique synthesized non-tryptic peptides from the ProteomeTools project were analyzed on a timsTOF-Pro to generate a dataset, **containing 93,227 MS/MS spectra of 74,847 unique peptides**, that was used to fine-tune an existing Prosit model.

Reviewer 2

Revision summary

Adams et al. adapted the existing Prosit model to predict peptide MS/MS fragmentation data, using timsTOF data and implemented it in Oktoberfest to rescore the MaxQuant PSMs. They prove that the (tims)TOF (CID) Prosit model outperforms the HCD- (Orbitrap)-Prosit model for predicting timsTOF MS/MS peptide spectra. Then, they show the benefits of this tool on previously published HLA class I and class II ligandomics (immunopeptidomics) data, increasing by up to 3-fold more unique peptides, depending on the dataset. In addition, the authors show that rescoring enhances the identification of non-canonical immunopeptides from nuORFs.

The authors clearly show the performance and benefits of their approach and also mention some of the limitations. The main innovation is adapting an MS/MS prediction model for tryptic and non-tryptic peptides based on data acquired from synthetic peptides, which can be considered a “ground truth.” Given the importance of MS-based immunopeptidomics for discovering therapy targets and the increasing usage of timsTOF instruments, this work is of high value to the community. However, some aspects require further evaluation, such as the effect of peptide size and type on the prediction and rescoring performance. I am confident that the authors will be able to address these issues. Thus, I recommend the publication of this manuscript after major revisions. Please find my comments and suggestions below.

We thank the reviewer for the constructive words. We performed more analyses to further evaluate the prediction and rescoring performances.

Comments to the authors Abstract

1. The following phrase seems incomplete. “Because the generation of immunopeptides from their parent proteins does not adhere to clear-cut rules, rather than being able to use known digestion patterns, every possible protein subsequence within HLA class-specific length restrictions needs to be considered” - into the database used for peptide identification?

This is a valid point. To make the phrase more complete we added “during database searching”:

Because the generation of immunopeptides from their parent proteins does not adhere to clear-cut rules, rather than being able to use known digestion patterns, every possible protein subsequence within HLA class-specific length restrictions needs to be considered during database searching.

2. It would be more informative to specify the number of acquired peptide spectra used for the model instead of or in addition to the total synthetic peptides. Please include it here.

This was also brought up by reviewer 1. To better represent the training data, we reported the number of confidently identified spectra and the number of unique peptide sequences acquired to train the model. We adjusted the text accordingly:

Over 300,000 synthesized non-tryptic peptides from the ProteomeTools project were analyzed on a timsTOF-Pro to generate a dataset, containing 93,227 MS/MS spectra of 73,325 unique peptides, that was used to fine-tune an existing Prosit model.

Introduction

3. Paragraph 2. The value of including mutations, pathogen genomes, and nuORFs in the database for immunopeptide identification needs to be clarified here. Similarly, the importance of immunopeptides for immunotherapy is mentioned in the abstract, introduction, and discussion, but it is not explained why or how are they used. Although these aspects would be clear for experts in immunopeptidomics, they may not be for other readers. Thus, please briefly explain these two points.

These are two very good suggestions that help to clarify the need for large search spaces in immunopeptidomics and emphasize the importance of this research field. Because we aim to make our results accessible to readers that are less familiar with immunopeptidomics, we implemented these suggestions the following ways:

The identification of immunopeptides is critical for the development of immunotherapies and vaccines. In recent years MS-based immunopeptidomics has been used to discover T cell targets against tumors, autoimmune diseases, and pathogens²⁻⁵. The identification of these targets is important for the development of immunotherapies, including the development of personalized vaccines and adoptive T cell transfers⁶.

In immunopeptidomics the search space is often expanded further by incorporating somatic mutations, pathogen genomes, and novel unannotated open reading frames (nuORFs). This will enable the detection of immunopeptides originating from these sources. A recent study highlights the significance of nuORFs as an underexplored source of MHC-I-presented, tumor-specific peptides that hold potential as targets for immunotherapy⁸.

Results general remarks

4. The authors named the new model "TOF Prosit 2023," but I recommend renaming it "timsTOF Prosit 2023" to represent better the data used for training. Indeed, the performance of the model on data from other TOF instruments was not tested. Moreover, it has been observed in other instruments that the ion mobility step may influence peptide fragmentation (ref. 30), and it is possible that it also happens in the timsTOF, as correctly mentioned by the authors in the Discussion paragraph 3.

We completely agree with the arguments for changing the name of the to "timsTOF Prosit 2023". We adjusted the text, figures, and publicly available model accordingly.

5. Peptide-HLA immunoprecipitated samples may contain peptides that are not necessarily HLA ligands. For instance, the antibody and HLA proteins may degrade into smaller peptides or other peptides may be co-enriched due to unspecific binding. Thus, I recommend adapting the labels of the figures (Fig. 3, Fig. 4a) and the text indicating the total number of peptides to “peptides” and using the terms “HLA-I ligands” and “HLA-II ligands” only for peptides that have been predicted to bind the correspondent HLA ligands. Although the prediction algorithms are imperfect, this provides a fairer comparison across the literature.

Following your suggestion, we have adapted figure 3 and 4a. To keep a clear separation between experiments where HLA-I-bound peptides are isolated compared to where HLA-II-bound peptides are isolated, “HLA-I ligands” was changed to “HLA-I peptides”.

6. Tryptic peptides were also included in the training and test datasets. Although the performance on the test dataset was not evaluated in function of peptide types, the results indicate that the model could be used for both tryptic and non-tryptic peptides. Thus, it would be valuable to show if the identification of this type of peptide is also improved by rescoring with the timsTOF Prosit model. I suggest a dataset with at least two different proteases to support the discussion (“... our model holds promise for numerous other biological and biomedical applications. One such area is deep proteome sequencing, where multiple proteases are used to enhance proteomic coverage”). It has been shown that immunopeptidomics MS/MS models may not translate so well to non-tryptic proteases and vice versa (Ref. 10, Declercq et al. 2021). Alternatively, a mixed proteome dataset could be used for validation since rescoring-rescued peptides should be associated with the corresponding species.

This is an excellent suggestion. By adding this we are able to highlight our model's usefulness beyond immunopeptidomics.

We found a dataset where DDA PASEF timsTOF Pro was used on a protein sample where for each sample a different protease was used. We were able to evaluate the usefulness of using rescoring on tryptic samples or samples where other proteases are used. The results were included in the manuscript:

Hence, PSM rescoring of timsTOF data drastically improved the identification rate of immunopeptides. To further evaluate how PSM rescoring performs when different proteases are used, we performed a reanalysis of samples cleaved with either trypsin, AspN, or GluC²⁷. These three proteases have distinct cleavage sites, with trypsin cleaving at the C-terminal side of lysine and arginine residues²⁸, AspN mainly cleaving at the N-terminal side of aspartic acid residues²⁹, and GluC mainly cleaving at the C-terminal side of glutamic acid residues²⁹.

PSM rescoring of the samples cleaved with AspN, GluC, and trypsin resulted in 1.5-fold, 1.7-fold, and 1.4-fold increases in unique identified peptides, respectively (Supplementary Fig. S7a). The performance of the timsTOF Prosit 2023 model was

stable across all proteases, with a median spectral angle of 0.77, 0.78, and 0.80 for samples cleaved with AspN, GluC, and trypsin, respectively (Supplementary Fig. S7b).

Results section 1

7. The authors analyzed an impressive number of 300,000 synthetic non-tryptic peptides, but only 93,227 were identified. (a) Can the authors explain why the identification rate was so low on the timsTOF instrument (31%), whereas the previous report (ref. 11) mentioned a synthesis/detection success ratio of 88%? (b) To better understand the nature of the peptides actually used for the model, a descriptive analysis should be included, at least in the supplementary material. For instance, the percentage of HLA class I binders, HLA class II binders, and AspN/LysN peptides; length distribution; and HLA allele coverage, e.g., against the twelve HLA supertype representatives from NetMHCpan 4.1 (<https://services.healthtech.dtu.dk/services/NetMHCpan-4.1/>).

302,105 unique synthesized non-tryptic peptides were measured, resulting in 360,448 MS/MS spectra. As mentioned in the methods, PSMs were filtered based on a PEP < 0.01 and an Andromeda score ≥ 70 , resulting in 93,227 MS/MS spectra.

A supplementary figure was added to show how the peptide types and peptide lengths are distributed across the training, validation, and test set:

In addition we have evaluated the HLA allele coverage against the twelve HLA supertype representatives from NetMHCpan 4.1 for each set.

8. Paragraph 2. Please mention here that the tryptic MS/MS spectra from ref. 2 were acquired using synthetic peptides. It was not clear to me until I looked for the reference.

We have adjusted the text accordingly:

Consequently, we compiled a dataset consisting of 93,227 non-tryptic MS/MS spectra, complemented by 184,552 previously published tryptic MS/MS spectra from 138,227 synthetic tryptic peptides²².

9. Fig. 1. The spectra showed many high-intensity peaks not assigned to the peptide GVDAANSAAQY, particularly on the Orbitrap side. This indicates that it could be a chimeric spectrum resulting from peptide coelution and co-isolation. Can the authors provide a cleaner example?

This peptide was randomly selected. In contrast to the timsTOF data, where there was only one spectrum found for this peptide, there were multiple spectra found in the Orbitrap data for this peptide. The Orbitrap spectrum was selected by comparing the spectral angle between all Orbitrap spectra and selecting the spectrum that is the

most similar. To include a cleaner example, another random peptide was selected resulting in the following spectra:

Fig. 1 Mirror plot of the singly charged non-tryptic synthetic ProteomeTools¹⁹ peptide VEDPVTVEY measured on a timsTOF (top; mzspect:PXD043844:HLAI_p2_97_178_p2-D1_S1-D1_1_6866.mgf:index:2341:VEDPVTVEY/1) and on an Orbitrap (bottom; mzspect:PXD021013:02446d_GD1-TUM_HLA_133_01_01-3xHCD-1h-R4:scan:32024:VEDPVTVEY/1) instrument. The spectral similarity measured by the normalized spectral contrast angle based on annotated fragments between the two spectra is 0.68. This illustrates how different timsTOF MS/MS spectra can look compared to Orbitrap data. This randomly chosen peptide was measured several times on the Orbitrap, after which the spectrum with the highest similarity to all other Orbitrap spectra for this peptide was selected (the medoid spectrum). In the timsTOF data the displayed MS/MS spectrum was the only measurement of this peptide. SA = normalized spectral contrast angle.

10. S1. Is the collision energy-Orbitrap in absolute values or normalized? Please specify it and add the units for both axes.

The collision energies are reported in absolute values. We have added the units to the axes:

Results section 2

11. Saying “277,781 MS/MS spectra obtained in this study” is misleading since the tryptic peptides were not acquired in this study. I think “compiled” would be a better word than “obtained.”

This is a valid point. We adjusted the text accordingly:

To optimize the Prosit fragment ion intensity prediction model towards timsTOF instruments, we fine-tuned the HCD Prosit 2020 model using the 277,779 MS/MS spectra compiled in this study, split into training, validation, and test sets (Fig. 2a).

12. (a) Which metric is used to conclude that “The TOF Prosit 2023 model demonstrates consistent performance across different precursor charges, peptide lengths, and collision energies (Supplementary Fig. S2a-c)?” The SA distribution differs slightly between charges (S2a). In addition, it decreases and becomes more spread in function of the peptide length. (b) Can the authors please elaborate on the reason for this behavior? (c) This should also be corrected in the footnote of Figure S2.

To evaluate whether the timsTOF Prosit 2023 model demonstrates consistency across different precursor charges, the test data were split into subsets according to the precursor charge. Then a Pearson correlation was calculated to evaluate whether there is a significant correlation between the spectral angle and the precursor charges. This resulted in a statistically significant correlation with a correlation coefficient of -0.19.

A similar analysis was performed to evaluate the model's consistency across aligned collision energies. This resulted in a statistically significant correlation with a correlation coefficient of 0.22.

In addition, we evaluated whether there was a significant correlation between the spectral angle and the length of the peptide. This resulted in a statistically significant correlation with a correlation coefficient of -0.38.

The manuscript and the description of Supplementary Figure S3 were updated to include these results:

The **timsTOF** Prosit 2023 model demonstrates consistent performance across different precursor charges and collision energies, with only a minor influence of these factors on the predicted fragment ion intensities (Pearson correlation coefficients of -0.19 and 0.22 for the precursor charge and the collision energy, respectively; Supplementary Fig. S3a,c). We observed a moderate influence of the peptide length on the predicted fragment ion intensities indicating that accurate fragment ion intensity prediction is more challenging for longer peptides (Pearson correlation coefficient of -0.38; Supplementary Fig. S3b).

(c) with only a minor influence of these factors on the predicted fragment ion intensities (Pearson correlation coefficients of -0.19 and 0.22 for the precursor charge and the collision energy, respectively). The model did show a moderate influence of the peptide length (b) on the predicted fragment ion intensities (Pearson correlation coefficient of -0.38).

13. (a) Can the authors specify the proportion of peptides used for each dataset (training/validation/test)? This could be included in the text or as part of Fig 2. In addition, it seems that the proportion of singly charged peptides in the validation and test datasets is lower than in the training (about 40%). (b) Is this just a visual effect, or was the proportion of singly charged so low? Could this affect the model? It is essential to ensure the performance of the model for singly charged peptides, considering that they may represent an important part of the HLA-I ligandome in some cases (Refs. 14, 15, 23).

Figure 2a. was updated to include values on top of the bars. This provides a better representation of the dataset. The proportion of singly charged peptides is 23% in the validation set, 25% in the test set, and 27% in the training set. We do not expect these slight differences in proportion to affect the model.

Results section 3

14. The increase in identifications is considerably smaller for peptides in the HLA-II dataset for both the Orbitrap and timsTOF. This is a limitation that is worth more attention. (a) Can the authors explain the possible reasons for the lower performance of Class II peptides? (b) Could it be derived from the peptide length-dependent effect observed in Fig. S2? (see point 12).

It is indeed true that longer peptides are more difficult to predict by the model. However, we think a larger effect of the lower performance of PSM rescoring on HLA-II peptides lies in that the Andromeda score will be higher for longer peptides. In addition,

there are less long decoys, making it easier to identify long peptides and separate them from incorrect shorter peptides. As a result, there are less PSMs to rescue through PSM rescoring.

Results section 4

15. The experiment with different sample amounts performed by Phulphagar et al. [30] was done by injecting 1 to 40 million A-375 cell equivalents from a larger sample. To avoid confusing this with a true initial sample amount, the term “cells” should be changed to “cell equivalents” in this section.

Thank you for bringing this to our attention. We adjusted the text and figures accordingly.

16. The authors mention: “The benefit of rescoring is expected to be even higher for low input samples. To validate this assumption, we performed a reanalysis on this timsTOF. To validate this assumption, we performed a reanalysis on this timsTOF SCP dataset using the TOF Prosit 2023 model.” I would have assumed the same. However, the increase was lower for 1 million (1.3-fold) than 40 million cell equivalents (1.9). Could the authors comment on the possible reasons for this?

This was also brought up by reviewer 1. We evaluated why our results did not support our hypothesis and discovered that there was a mistake in our initial hypothesis. It stated that low sample inputs frequently suffer from missing peaks, as fragments with low intensities fail to surpass the noise level and that this leads to low database search engine scores and low identification rates. However, as we mention in the introduction, MS/MS spectra from timsTOF instruments exhibit more reproducibility at low abundances compared to MS/MS spectra from Orbitrap instruments. Consequently, we do not necessarily expect low database search engine scores and low identification rates.

Our explanation on why there is a lower increase for 1 million cell equivalents is that MaxQuant was already quite capable of annotating the spectra, with an identification rate of ~50%. After PSM rescoring an identification rate of almost 70% is reached. As a result, not a lot of MS/MS spectra were left to annotate.

Discussion

17. Although the dataset used for the model was very large, it is not genuinely comprehensive (“including all or nearly all elements or aspects of something”). I would recommend using the word “extensive” instead of “comprehensive” in the phrase “[...] we established a comprehensive dataset [...]”

The text was changes accordingly:

In this study, we established an **extensive** dataset consisting of MS/MS spectra from synthetic non-tryptic and tryptic peptides measured on a timsTOF instrument.

18. “Similarly to what has been observed for retention time alignment 31, we expect a benefit from collision energy alignment to account for the run-to-run fluctuations.” Is this something that was addressed in this manuscript or something that the authors plan to implement in the future? This was not clear to me; please clarify it.

We appreciate that you bring this to our attention. The CE calibration is very important for accurate fragment ion intensity predictions. We wish this to be clear in the manuscript. To achieve this we added a paragraph in the second section of the results:

It is important to note that the applied collision energy has a profound impact on the information content of the obtained MS/MS spectra ²⁴. To optimize the transfer learning, the collision energies of the training, validation, and test set were calibrated according to the collision energies the HCD Prosit 2020 model would expect. To achieve this, a robust linear model was trained on the entire training set for each precursor charge state and for tryptic and non-tryptic peptides separately. The

acquired linear models were applied on the training, validation, and test set. To ensure accurate fragment ion intensity predictions, collision energy calibration is needed.

Methods

19. Data acquisition / Synthetic non-tryptic peptides data acquisition / paragraph 2. (a) There seems to be an error in the description of the TIMS range used for the acquisition of non-tryptic peptide data. I downloaded some of the files and the range is 0.7 to 1.65 Vs/cm² and not 0.7 to 1.645 as written in the methodology. Please correct the text. (b) In addition, the authors should mention that no isolation polygon was used to select precursors for MS/MS. Otherwise, one may assume that the standard polygon for proteomics was used, which would result in the exclusion of singly charged peptides. (c) Besides, some of the larger singly charged peptides have an inversed ion mobility above 1.65 Vs/cm² (See ref. 15). Thus, a TIMS range of 0.6 to 1.8 Vs/cm² could have resulted in more identifications. The authors should comment on this as a limitation in the Discussion.

The reviewer is correct that the end range of the TIMS was mistakenly reported at 1.45. The TIMS range starts at 0.70 Vs/cm² and ends at 1.70 Vs/cm². We added in the text that indeed no polygon was used, so singly charged ions are included. Ref. 15 contains many TimsTOF optimization steps that cumulatively improve the yield of HLA peptide identification. The TIMS range they tested was 0.20 Vs/cm² - 1.60 Vs/cm² and 0.60 Vs/cm² - 1.60 Vs/cm², the latter resulted in an increase of identified peptides compared to previously reported ranges up to 1.30 Vs/cm². The range used in this article was proposed by Bruker in personal correspondence. Of course, we cannot exclude that additional peptides are identified if the upper TIMS range is further increased, although data presented in figure 1 from Ref. 15 (<https://www.nature.com/articles/s41467-023-42692-7>) and Figure 2 or Ref. 16 (<https://www.researchsquare.com/article/rs-2625909/v1>) suggest that the bulk of the peptides should be identified below 1.70 Vs/cm². The Thunder method described in Ref 16. does go to 1.75 Vs/cm², and we would have broadened the range if it was published before the onset of our measurements.

The TIMS-MS survey scan was acquired between 0.70 and 1.70 Vs/cm² and 100–1,700 m/z with a ramp time of 100 ms. The m/z and ion mobility information was used to select precursors with charges ranging from 1–3. No polygon was used for precursor ion selection.

20. a) Were the cysteine-containing peptides synthesized with carbamidomethylated cysteine, or were they treated to enforce this modification? Otherwise, why was carbamidomethylated cysteine used as a fixed modification? b) Contrary to proteomics workflows, immunopeptidomics preparation typically does not include the reduction and alkylation steps. Thus, using carbamidomethylated cysteine may negatively affect the detection of cysteine-containing HLA peptides (<https://doi.org/10.1038/s42003-021-02364-y>). The authors should evaluate this.

This is a very relevant comment, because there is a known underrepresentation of cysteine in immunopeptidomics datasets and we did not mention this aspect in the manuscript. a) Yes, during peptide synthesis, carboxyamidomethylated cysteine building blocks were used to eliminate the need for cysteine modification before MS analysis (<https://www.nature.com/articles/nmeth.4153>). b) In some immunopeptidomics protocols, cysteines are not carbamidomethylated. However, in other protocols they are. For example, in the 'SCP study' iodoacetamide was used during the sample preparation, whereas in the 'comparison study' cysteines were not carbamidomethylated.

In the initial manuscript each dataset was analyzed with MaxQuant with cysteine carbamidomethylation as a fixed modification. However, for the 'comparison dataset' these settings are incorrect, because there should not be any carbamidomethylated cysteine in the dataset. To correct this mistake, we first reprocessed the entire 'comparison dataset' and searched it with MaxQuant without cysteine carbamidomethylation as a fixed modification. Then the entire dataset was also again rescored with Percolator through oktoberfest. However, because the current Prosit models are not trained on peptides containing free cysteine side chains, 8% of potential target PSMs were lost because they could not be rescored. This is also mentioned in the manuscript:

Because the current Prosit models were not trained on peptides containing free cysteine side chains or other amino acid modifications that may be identified on immunopeptides, 8% of potential target PSMs (of which 99% would not survive the $PEP < 0.01$ filter) were lost because they could not be rescored.

The methods were updated accordingly:

Carbamidomethylated cysteine was not included as a fixed modification, because cysteine was not carbamidomethylated during sample processing. Before PSM rescoring all PSMs containing free cysteine side chains were removed.

Other comments

21. Please add page and line numbers when resubmitting. It helps reviewers make comments on specific lines.

Page and line numbers were added for the resubmission.

22. Define non-standard abbreviations the first time they are used. For instance, "SPD".

We agree that defining abbreviations on their first use is essential to prevent complex jargon. We have carefully re-read the full manuscript to ensure that all acronyms have been properly defined.

23. Some preprints are cited across the manuscript, which I see as a positive point. However, the references should specify their preprint status. For instance, this is

correctly mentioned in references 12 and 39 but not in references 14 and 15. In addition, MSBooster was published during the revision time; please replace the preprint with the final reference <https://doi.org/10.1038/s41467-023-40129-9>.

Since submitting the initial manuscript a lot of the preprints we initially cited, got published. We updated all the references.

24. I downloaded some of the files from the dataset PXD043844. When I unzip them, the raw file is borrowed in a series of folders. This may create problems when others try to repurpose the data. Please correct this if it is still possible. (e.g., HLA1_1_96_p2-A4_S2-A4_1_6929.d\Users\adams\Projects\non-tryptic-raw\HLA1_1_96_p2-A4_S2-A4_1_6929.d).

Thank you for bringing this to our attention. We verified this for all the datasets we put online to avoid this issue and fixed it where necessary.

Reviewer 3

This work's primary contribution is the generation of a large number of synthetic peptide spectra using the TimsTOF-Pro instrument. The data set could be used to fine-tune the ProSight model for fragment ion intensity prediction. The authors demonstrated that PSM rescoring using the fine-tuned ProSight could greatly improve the identification of immunopeptides. The data set of peptide spectra generated in this study must be an invaluable public resource and would be used in numerous studies. However, the methods presented by the authors are not novel; fragment ion prediction-based rescorings have been utilized in numerous studies. This study appears more appropriate for a data resource paper than a research article. To demonstrate utility, additional results should be provided.

We agree that the dataset generated in this study is extremely valuable. To increase its reusability and to be more transparent, we added more information in the manuscript:

Consequently, we compiled a dataset consisting of 93,227 non-tryptic MS/MS spectra, complemented by 184,552 previously published tryptic MS/MS spectra from 138,227 unique synthetic tryptic peptides²². This extensive dataset, comprising a total of 277,779 MS/MS spectra and 216,934 unique peptides, serves as a unique training dataset for the development of machine learning tools tailored to timsTOF instruments (Supplementary Fig. S2).

A reanalysis of a dataset where multiple proteases were used was performed illustrating the utility of our model.

Hence, PSM rescoring of timsTOF data drastically improved the identification rate of immunopeptides. To further evaluate how PSM rescoring performs when different proteases are used, we performed a reanalysis of samples cleaved with either trypsin, AspN, or GluC²⁷. These three proteases have distinct cleavage sites, with trypsin cleaving at the C-terminal side of lysine and arginine residues²⁸, AspN mainly cleaving at the N-terminal side of aspartic acid residues²⁹, and GluC mainly cleaving at the C-terminal side of glutamic acid residues²⁹.

PSM rescoring of the samples cleaved with AspN, GluC, and trypsin resulted in 1.5-fold, 1.7-fold, and 1.4-fold increases in unique identified peptides, respectively (Supplementary Fig. S7a). The performance of the timsTOF Prosit 2023 model was stable across all proteases, with a median spectral angle of 0.77, 0.78, and 0.80 for samples cleaved with AspN, GluC, and trypsin, respectively (Supplementary Fig. S7b).

Comments:

- Figure 3 depicts the identification improvement resulting from rescoring based on fragment ion prediction. How did the authors identify the initial peptides for the comparison? In calculating FDR for the initial peptide identification, did the authors employ Andromeda scores without any rescoring? PSM rescoring (using Percolator, for example) has become widespread. The fragment ion prediction is not necessary for rescoring, which takes into account multiple other factors. Therefore, the improvement in peptide identification should be demonstrated by comparing rescoring results without and with fragment ion prediction. This comparison will clearly show the impact of fragment ion prediction. Figure 3 should also include the results of timsTOF - CID Prosit 2020, which will demonstrate the usefulness of the fine-tuned model using a synthetic timsTOF data set.

This is a very valid comment. In the original submission we compared MaxQuant + Prosit + Percolator against MaxQuant + Percolator. However, this was unclear in the original manuscript and is adjusted for this version:

We compared results from MaxQuant + Percolator vs MaxQuant + Prosit + Percolator, respectively the feature sets can be found in Supplementary Table S1 and S2.

To investigate the impact of fragment ion intensity prediction, we performed MaxQuant + Prosit + Percolator again, however only with fragment ion intensity prediction based features, thus removing the retention time prediction based features. This was compared against MaxQuant + Percolator. We did this analysis with all the different models mentioned in this study: timsTOF Prosit 2023, HCD Prosit 2020, and CID Prosit 2020 model. We observed a consistently higher gain after rescoring if the timsTOF Prosit 2023 model was used to generate the spectral features. A paragraph was added to the results and two new panels were added to Figure 3:

To evaluate the effect of the fragment ion intensity prediction model, PSM rescoring was performed on the timsTOF data with the different models. To isolate the benefit of fragment ion intensity information during rescoring, RT prediction-based features were excluded during this analysis (Supplementary Table S1). We observed that PSM rescoring with the timsTOF 2023 model consistently resulted in a higher gain compared to PSM rescoring with the other two models (Fig. 3e-f). As expected the timsTOF Prosit 2023 model results in a higher gain of identifications compared to the HCD Prosit 2020 and the CID Prosit 2020 model. A possible explanation as to why PSM rescoring with the HCD Prosit 2020 model consistently results in a higher increase compared to PSM rescoring with the CID Prosit 2020 model, is that both HCD and timsTOF have a beam-type fragmentation and are thus more similar compared to CID.

Figure 3. Gained, shared, and lost peptide identifications for different sample types to compare PSM rescoring on Orbitrap data with PSM rescoring on timsTOF data. In general PSM rescoring was able to boost the confidence in peptide identifications, retaining true PSMs, gaining new PSMs, and losing only a small number of previously incorrect PSMs. **a** On average PSM rescoring of HLA-I Orbitrap data with the CID Prosit 2020 model resulted in a 2.2-fold increase (with 27,023 shared, 28,890 gained, and 486 lost unique peptides). **b** On average PSM rescoring of HLA-I timsTOF data with the timsTOF Prosit 2023 model resulted in a 2.7-fold increase (with 41,822 shared, 62,962 gained, and 521 lost unique peptides). **c** On average PSM rescoring of HLA-II Orbitrap data with the HCD Prosit 2020 model resulted in a 1.4-fold increase (with 30,081 shared, 12,187 gained, and 537 lost unique peptides). **d** On average PSM rescoring of HLA-II timsTOF data with the timsTOF Prosit 2023 model resulted in a 1.8-fold increase (with 47,201 shared, 26,852 gained, and 787 lost unique peptides). **e** To evaluate the effect of the fragment ion intensity prediction model on PSM rescoring, the RT prediction-based features were excluded. On average PSM rescoring of HLA-I timsTOF data with the timsTOF Prosit 2023 model resulted in a 2.8-fold increase (with 39,606 shared, 59,458 gained, and 549 lost unique peptides). On average PSM rescoring of HLA-I timsTOF data with the HCD Prosit 2020 model resulted in a 2.5-fold increase (with 39,296 shared, 52,520 gained, and 859 lost unique peptides). On average PSM rescoring of HLA-I timsTOF data with the CID Prosit 2020 model resulted in a 2.2-fold increase (with 38,431 shared, 43,748 gained, and 1,724 lost unique peptides). **f** To evaluate the effect of the fragment ion intensity prediction model on PSM rescoring, the RT prediction-based features were excluded. On average PSM rescoring of HLA-II timsTOF data with the timsTOF Prosit 2023 model resulted in a 1.5-fold increase (with 46,439 shared, 25,443 gained, and 918 lost unique peptides). On average PSM rescoring of HLA-II timsTOF data with the HCD Prosit 2020 model resulted in a 1.5-fold increase (with 45,393 shared, 24,250 gained, and 964 lost unique peptides). On average PSM rescoring of HLA-II timsTOF data with the CID Prosit 2020 model resulted in a 1.5-fold increase (with 46,088 shared, 21,899 gained, and 1,269 lost unique peptides). RCC = renal cell carcinoma; HNSCC = head and neck squamous-cell carcinoma; PBMC = peripheral blood mononuclear cell; CLL = chronic lymphocytic leukemia.

Reviewers' Comments:

Reviewer #1:

Remarks to the Author:

After reading the authors' response and the revised manuscript I have no further comments. I recommend publishing the manuscript.

Adams et al. Fragment ion intensity prediction improves the identification rate of non-tryptic peptides in TimsTOF

Revision summary

In this revised version, Adams *et al.* provided an improved manuscript presenting the tuning of a Prosit model to predict peptide MS/MS fragmentation data for timsTOF instruments and its application for identifying immunopeptides. I thank the Authors for their work and detailed answers in the rebuttal document. The Authors correctly addressed most of the issues raised by the Reviewers. However, some aspects require further evaluation, such as the effect of using wide m/z tolerance thresholds. In addition, I have some minor recommendations to improve the clarity of the manuscript. Thus, I recommend the publication of this manuscript after revisions. Please find my comments and suggestions below.

Major comments

1. I think that a precursor tolerance of 40 ppm is too high for timsTOF instruments (L354, L426). The timsTOF Pro typically provides a resolution of 35,000 to 40,000 when operating correctly. Similarly, 20 ppm seems too high for Orbitrap instruments (L426). In contrast, the authors of the publication presenting the data employed for the comparison (Ref. 15) used precursor tolerances of 20 and 5 ppm for the timsTOF and Orbitrap data, respectively. Such large thresholds could have an important effect on the results. For instance, more decoys or false positives could be identified. This could be negatively impacting the FDR control and, thus, peptide identification (see, for example, https://www.agilent.com/cag/EMEA/Pharma_Tour_presentations/06-High_Confidence_Protein.pdf). Can the authors comment on their reason for selecting such large thresholds? To verify if these thresholds didn't impact the result negatively, the authors should plot the mass accuracy vs. score of target- vs. decoy- peptides (e.g., with a different color). Alternatively, or in addition, plotting the distribution of target and decoy peptides in function of the mass accuracy could provide similar information. If the large m/z tolerance does not negatively impact the results, the authors should at least mention their rationale in the manuscript and highlight that stricter thresholds are typically used. However, if the results are negatively impacted, the authors may consider reanalyzing the data with narrower thresholds.
2. Also, was the fragment tolerance similar or even higher? Please mention it in the methods. If it was higher than 20 ppm (or 0.03 Th) for any of the instruments, the authors should justify and evaluate their choice as suggested for the precursor tolerance.

3. L104. I am still surprised by the low identification rate since only 93,227 (31%) out of 302,105 synthetic non-tryptic peptides were identified in the timsTOF analyses, while 88% were identified in ref. 12 using an Orbitrap Fusion Lumos MS. However, previous studies have shown a higher coverage of the immunopeptidome in timsTOF instruments than Orbitrap instruments (ref. 15), albeit it was an Exploris 480. In the rebuttal document, the authors explained the low coverage by referring to the thresholds used to filter PSMs (PEP < 0.01 and an Andromeda score ≥ 70), but I suspect the low identification rate could be due to non-optimal acquisition settings and/or the large mass tolerance mentioned in Comment 1. The authors thoroughly described the peptides identified, showing a good distribution across the peptide charges, types, and HLA binding motifs. However, it would have been valuable to compare them to the peptides that were not identified, which could indicate if the acquisition method created a bias towards or against specific type of peptides. Since deeply evaluating the performance of the acquisition method could be beyond the scope of the manuscript, I suggest that the authors mention the low identification rate as a limitation in the discussion and point out that coverage could have been improved using optimized methods (refs. 15, 16, 30).

Minor comments

Abstract and Introduction

4. L20-21: "Low abundant peptides often occur in" immunopeptidomics samples/analyses (not in the field of immunopeptidomics) – please correct
5. L36-37: Although the field of MS-based immunopeptidomics seems to be currently booming, the first studies date from the 90s. Thus, the phrase "in recent years" doesn't apply here. Please remove it.

Results

6. L98. "...all measurements conducted in previous studies were performed on Orbitrap and ion trap instruments". This is not entirely true since the data of the ProteomeTools tryptic peptides was acquired in timsTOF Pro in ref. 22. Please correct it.
7. L123-130: I think that a Pearson correlation is not the proper method to assess the association between a continuous variable (the spectral angle) and a discrete nominal variable with only three (the charge). In this case, a simple median test would be more informative.
8. L130-131. This phrase sounds ambiguous or incomplete. Did the authors mean there was a bias in function of the C- and N-terminal residue and not "towards"? There seems to be a bias in function of the residue in the terminal positions, but I didn't see a figure showing a better SA for amino acids in these positions. Please correct it and briefly mention what the bias is. Mainly, it is worth commenting that peptides with K or R at the C-terminus have a higher SA, indicating a bias towards tryptic peptides, probably due to the large proportion of such peptides in the training dataset. Please comment on whether it could bias immunopeptide identification in the function of their sequence motifs.

9. Figure 3. The figure legend is difficult to read with all the numeric values described in parentheses. Although I understand that the authors added the values to show the low number of lost identifications, I recommend presenting the values in a supplementary table or dataset (Excel or CSV) and summarizing the figure description.
10. L176-177. "Drastically" sounds rather exaggerated. Please use a less sensational word. In addition, this phrase seems to belong to the previous paragraph. The authors may consider moving it there.

Discussion

11. L247-248. Although one could expect an improvement for other search engines, this was not tested here. In addition, the improvement may not be as high as it is for MaxQuant. Indeed, several publications have shown that other search engines outperform MaxQuant for immunopeptidomics search without PSM rescoring. In addition, when reading this phrase, one may think that the timsTOF model is ready to be applied for rescoring results from other engines. However, the last paragraph mentions that it is not. To better link these observations, please move the phrase "Based on previous studies [11,12,36]; we expect improvement when applied to other search engines as well." to the last paragraph (e.g., at the beginning). This would also help with the narrative by using the first paragraph to summarize the results, and the last one to talk about the limited use for MaxQuant and some perspectives.
12. Not including peptides with free and cysteinylated cysteines is an important limitation of the current ProSight models for identifying immunopeptides. Similarly, dropping Cys-containing peptides during rescoring is an important limitation of the workflow presented in this study. Indeed, it has been shown that such peptides can represent an important part of the HLA-ligandome (see, e.g., ref. 39; and Bauza-Martinez, 2021). Although the authors mention it in the results, I think it is worth briefly mentioning this limitation in the discussion (e.g., in the second paragraph).
13. L260. the authors may mean that "there is a bias [in function] of the C-terminal arginine or lysine residues" and not "towards" them. See also my comment in the Results section (Comment 8). Please check and correct if applicable.
14. L276. Reference 40 was also trained on tryptic peptides and phosphopeptides for CCS prediction. It's worth mentioning it here for clarity.
15. L277-285. I still think that this paragraph seems to indicate using collision energy calibration as a perspective and not a result. However, the authors added the explanation of the collision energy calibration to the Results section and highlighted its importance in the rebuttal document. In addition, the Oktoberfest manuscript mentions that it performs collision energy calibration. Thus, the authors may consider mentioning it at the end of the paragraph. For instance: "Similarly to what has been observed for retention time alignment [42], we expect a benefit from collision energy alignment to account for the run-to-run fluctuations. **Therefore, the collision energy calibration is implemented in Oktoberfest for PSM rescoring [26].**"

Data availability

16. I downloaded again some of the files from the MassIVE dataset MSV000092456. When I unzip them, the raw folder (.d) is still borrowed in a series of folders. However, the authors said they corrected this. I may be accessing an old version, but the authors should make sure this is corrected for their final submission. For instance, “[my_folder]\Users\adams\Projects\non-tryptic-raw\HLAI_1_96_p2-H5_S2-H5_1_7014.d”.

Multiple proteases experiment

Thank you for testing the performance of the timsTOF Prosit model on multiple proteases. It adds considerable value to the manuscript since it shows that the model can also be used for deep proteomics profiling. Thus, I have a few suggestions to highlight this further. However, I would understand if the authors prefer not to implement these and focus only on telling the story about improving immunopeptidomics experiments.

17. I suggest also mentioning the improvement in identifying peptides derived from proteomics experiments when the results are summarized in the Abstract, Introduction (last paragraph), and Discussion (first paragraph).
18. L177-186. To facilitate finding these results when reading the subtitles, the results could be presented in an additional subsection, independently of the immunopeptidomics experiments. In addition, Fig. S7b could be included in the main text instead of being borrowed.

Reviewer #3:

Remarks to the Author:

The authors have adequately addressed the comments I made in a previous review. I recommend that the authors consider a few points for improvement.

Comments:

1. The timsTOF and HCD spectra appear to be quite different (Figure 1). Nevertheless, it seems that there were no differences between using timsTOF and HCD models in rescoring HLA-II peptides from timsTOF data. Could the authors add some comments on the reason for this?

2. What is the difference between the data used in Figure 3 and Supplementary Figure S5? It is difficult to see the difference in the data. In addition, writing the numbers of PSMs in the legend is complicated (too many numbers).

3. Wasn't the data used in Supplementary Figure S6b Orbitrap-HCD? Yes, but it was referred to as timsTOF data in the legend. Also, please explicitly state Orbitrap CID for Figure S6a.

Response to the Review

Reviewer 1

After reading the authors' response and the revised manuscript I have no further comments. I recommend publishing the manuscript.

We thank the reviewer for taking the time to thoroughly review our manuscript and for their positive feedback. We greatly appreciate the recommendation to publish the manuscript.

Reviewer 2

In this revised version, Adams et al. provided an improved manuscript presenting the tuning of a Prosit model to predict peptide MS/MS fragmentation data for timsTOF instruments and its application for identifying immunopeptides. I thank the Authors for their work and detailed answers in the rebuttal document. The Authors correctly addressed most of the issues raised by the Reviewers. However, some aspects require further evaluation, such as the effect of using wide m/z tolerance thresholds. In addition, I have some minor recommendations to improve the clarity of the manuscript. Thus, I recommend the publication of this manuscript after revisions. Please find my comments and suggestions below.

We appreciate the reviewer's kind words and constructive comments. We have carefully reconsidered and revised the manuscript to enhance its clarity. Please find our responses to the specific comments provided below.

Major comments

1. I think that a precursor tolerance of 40 ppm is too high for timsTOF instruments (L354, L426). The timsTOF Pro typically provides a resolution of 35,000 to 40,000 when operating correctly. Similarly, 20 ppm seems too high for Orbitrap instruments (L426). In contrast, the authors of the publication presenting the data employed for the comparison (Ref. 15) used precursor tolerances of 20 and 5 ppm for the timsTOF and Orbitrap data, respectively. Such large thresholds could have an important effect on the results. For instance, more decoys or false positives could be identified. This could be negatively impacting the FDR control and, thus, peptide identification (see, for example, https://www.agilent.com/cag/EMEA/Pharma_Tour_presentations/06-High_Confidence_Protein.pdf). Can the authors comment on their reason for selecting such large thresholds? To verify if these thresholds didn't impact the result negatively, the authors should plot the mass accuracy vs. score of target- vs. decoy- peptides (e.g., with a different color). Alternatively, or in addition, plotting the distribution of target and decoy peptides in function of the mass accuracy could provide similar information. If the large m/z tolerance does not negatively impact the results, the authors should at least mention their rationale in the manuscript and highlight that stricter thresholds are typically used. However, if the results are negatively impacted, the authors may consider reanalyzing the data with narrower thresholds.

After carefully looking at the MaxQuant parameters, we noticed a mistake was made when reporting the precursor tolerance. The correct precursor tolerance is 10 ppm. The methods section was updated accordingly:

For timsTOF data the precursor mass tolerance was set to 10 ppm and the fragment mass tolerance was set to 40 ppm. For Orbitrap data the precursor mass tolerance was set to 4.5 ppm and the fragment mass tolerance was set to 20 ppm.

Also note that using very small precursor mass tolerances can potentially bias the target-decoy approach as an insufficient number of candidates might be left, leading

to an underestimation of the FDR (e.g. <https://pubs.acs.org/doi/full/10.1021/pr5007284>). In contrast, in our approach, as features derived from the precursor mass difference are included during Percolator rescoring, ensures that (i) a sufficient number of target and decoy candidates are available to ensure proper estimation of the FDR, and (ii) Percolator will easily learn to eliminate decoys and other false matches that exhibit an excessive precursor mass difference.

2. Also, was the fragment tolerance similar or even higher? Please mention it in the methods. If it was higher than 20 ppm (or 0.03 Th) for any of the instruments, the authors should justify and evaluate their choice as suggested for the precursor tolerance.

The fragment tolerance was 40 ppm, which is the default MaxQuant value for TOF instruments (<https://www.ncbi.nlm.nih.gov/pmc/articles/PMC7261821/>). We have added the fragment mass tolerance in the description of the used parameters.

For timsTOF data the precursor mass tolerance was set to 10 ppm and the fragment mass tolerance was set to 40 ppm. For Orbitrap data the precursor mass tolerance was set to 4.5 ppm and the fragment mass tolerance was set to 20 ppm.

3. L104. I am still surprised by the low identification rate since only 93,227 (31%) out of 302,105 synthetic non-tryptic peptides were identified in the timsTOF analyses, while 88% were identified in ref. 12 using an Orbitrap Fusion Lumos MS. However, previous studies have shown a higher coverage of the immunopeptidome in timsTOF instruments than Orbitrap instruments (ref. 15), albeit it was an Exploris 480. In the rebuttal document, the authors explained the low coverage by referring to the thresholds used to filter PSMs ($PEP < 0.01$ and an Andromeda score ≥ 70), but I suspect the low identification rate could be due to non-optimal acquisition settings and/or the large mass tolerance mentioned in Comment 1. The authors thoroughly described the peptides identified, showing a good distribution across the peptide charges, types, and HLA binding motifs. However, it would have been valuable to compare them to the peptides that were not identified, which could indicate if the acquisition method created a bias towards or against specific type of peptides. Since deeply evaluating the performance of the acquisition method could be beyond the scope of the manuscript, I suggest that the authors mention the low identification rate as a limitation in the discussion and point out that coverage could have been improved using optimized methods (refs. 15, 16, 30).

Given the adage "garbage in = garbage out" when training machine learning models, we explicitly used stringent thresholds for filtering the PSMs to ensure that only the highest quality PSMs were used for training. Nevertheless, the mismatch in identification rate compared to the original Orbitrap analyses of the synthesized libraries is indeed somewhat surprising. This can be due to several reasons, however, ranging from library redistribution, acquisition, search engine, and software versions among others. We agree with the reviewer that protocols could be optimized, however

as this is out of the scope of the current manuscript, we now mention this caveat in the Discussion as follows:

While not all synthesized peptides that are theoretically present could be identified, which could be due to both factors related to the acquisition method and the bioinformatics analyses, this enabled us to construct a valuable ground truth dataset that served as the foundation for training the novel timsTOF Prosit 2023 model.

Minor comments

Abstract and Introduction

4. L20-21: “Low abundant peptides often occur in” immunopeptidomics samples/analyses (not in the field of immunopeptidomics) – please correct

The sentence was adjusted accordingly:

Low abundant peptides often occur in immunopeptidomics experiments, which is why the highly sensitive timsTOF instruments are increasingly gaining popularity.

5. L36-37: Although the field of MS-based immunopeptidomics seems to be currently booming, the first studies date from the 90s. Thus, the phrase “in recent years” doesn't apply here. Please remove it.

“In recent years” was removed.

Results

6. L98: “...all measurements conducted in previous studies were performed on Orbitrap and ion trap instruments”. This is not entirely true since the data of the ProteomeTools tryptic peptides was acquired in timsTOF Pro in ref. 22. Please correct it.

This refers to data used to train fragment ion intensity prediction models. The synthesized tryptic peptides were measured on a timsTOF Pro to train a CCS prediction model. We have adjusted this sentence to make this more clear:

However, all measurements conducted to train previous Prosit models were performed on Orbitrap and ion trap instruments.

7. L123-130: I think that a Pearson correlation is not the proper method to assess the association between a continuous variable (the spectral angle) and a discrete nominal variable with only three (the charge). In this case, a simple median test would be more informative.

We updated the statistical test to use a one-way ANOVA followed by Tukey's post hoc test. The one-way ANOVA test was significant, rejecting the null hypothesis that the mean model performance is the same across all three charge states. To investigate

this further, we performed Tukey's post hoc test. This showed that the difference between the mean performance of charge state 1 and 3 and charge state 2 and 3 was significant, whereas there was no significant difference in the model's performance between charge state 1 and 2. The result section was updated accordingly:

The timsTOF Prosit 2023 model demonstrates consistent performance across different precursor charges, with only a minor decrease of the performance for peptides with charge state 3 compared to charge state 1 and 2 (one-way ANOVA followed by Tukey's post hoc test; Supplementary Fig. S3a). We observed a moderate influence of the peptide length on the predicted fragment ion intensities, indicating that accurate fragment ion intensity prediction is more challenging for longer peptides (Pearson correlation coefficient of -0.38; Supplementary Fig. S3b).

In addition, we used the one-way ANOVA followed by Dunnett's post hoc test to investigate the bias in function of C- and N-terminal amino acids:

The timsTOF Prosit 2023 model demonstrates consistent performance across different collision energies, with only a minor influence on the predicted fragment ion intensities (Pearson correlation coefficient of 0.22 for the collision energy; Supplementary Fig. S3c). We observed a small bias in function of C- and N-terminal amino acids, specifically, the performance is higher for peptides with an arginine or a lysine at the C-terminus (one-way ANOVA followed by Dunnett's post hoc test, Supplementary Fig. S4a-b). This is likely a result of the large proportion of tryptic data in the training data (Supplementary Fig. S2a).

Furthermore, we have added a section in the Methods:

Statistical analysis

To evaluate the consistency of the timsTOF Prosit 2023 model across different peptide lengths and collision energies, a Pearson correlation was performed, resulting in Pearson correlation coefficients of -0.38 and 0.22, respectively. In addition, we investigated the influence of the charge state on the fragment ion intensities with a one-way ANOVA, resulting in a p-value of 6.97E-247. Subsequently, Tukey's post hoc test was used to assess the differences between charge states 1 and 2 (p-value = 0.96), 1 and 3 (p-value = 5.56E-12), and 2 and 3 (p-value = 5.56E-12). To evaluate the effect of N- and C-terminal amino acids on the model's performance, a one-way ANOVA was used, resulting in p-values 1.85E-27 and 3.68E-60, respectively. Then, Dunnett's post hoc test was used to compare peptides with a C-terminal arginine or lysine to all peptides (p-value = 4.95E-09). All statistical tests were performed on the test set.

8. L130-131. This phrase sounds ambiguous or incomplete. Did the authors mean there was a bias in function of the C- and N-terminal residue and not "towards"? There seems to be a bias in function of the residue in the terminal positions, but I didn't see a figure showing a better SA for amino acids in these positions. Please correct it and briefly mention what the bias is. Mainly, it is worth commenting that peptides with K or R at the C-terminus have a higher SA, indicating a bias towards tryptic peptides, probably due to the large proportion of such peptides in the training dataset. Please

comment on whether it could bias immunopeptide identification in the function of their sequence motifs.

We adjusted the sentence accordingly:

We observed a small bias in function of C- and N-terminal amino acids, specifically, the performance is higher for peptides with an arginine or a lysine at the C-terminus (one-way ANOVA followed by Dunnett post hoc analysis, Supplementary Fig. S4a-b). This is likely a result of the large proportion of tryptic data in the training data (Supplementary Fig. S2a).

To address the potential bias this could introduce, a few sentences were added in the discussion as well:

Although our model still exhibits a slight performance improvement for tryptic peptides, likely due to a larger number of such training samples, this did not preclude us from achieving strong performance for non-tryptic peptides.

9. Figure 3. The figure legend is difficult to read with all the numeric values described in parentheses. Although I understand that the authors added the values to show the low number of lost identifications, I recommend presenting the values in a supplementary table or dataset (Excel or CSV) and summarizing the figure description.

To make the figure legend more readable, all numeric values in the parentheses were removed. To ensure readers can find the values from the figures, we added a source data file containing all values from shared, gained, and lost plots. We referenced to the source data file in all relevant figures:

Source data are provided as a source data file.

10. L176-177. "Drastically" sounds rather exaggerated. Please use a less sensational word. In addition, this phrase seems to belong to the previous paragraph. The authors may consider moving it there.

After looking at the sentence again and considering moving it to the previous paragraph, we have decided to remove it from the manuscript.

Discussion

11. L247-248. Although one could expect an improvement for other search engines, this was not tested here. In addition, the improvement may not be as high as it is for MaxQuant. Indeed, several publications have shown that other search engines outperform MaxQuant for immunopeptidomics search without PSM rescoring. In addition, when reading this phrase, one may think that the timsTOF model is ready to be applied for rescoring results from other engines. However, the last paragraph mentions that it is not. To better link these observations, please move the phrase "Based on previous studies [11,12,36]; we expect improvement when applied to other

search engines as well." to the last paragraph (e.g., at the beginning). This would also help with the narrative by using the first paragraph to summarize the results, and the last one to talk about the limited use for MaxQuant and some perspectives.

The paragraph was adjusted accordingly:

Although currently the timsTOF Prosit 2023 model is dependent on MaxQuant, as it relies on the search engine to sum the MS/MS scans, in the future, it will be further extended to support other search engines as well and become search engine agnostic, similar to how Oktoberfest has recently extended applicability of Prosit Orbitrap predictions beyond MaxQuant. **Based on previous studies^{11,12,36}, we expect an improvement of the identification rate when applied to other search engines as well.** The timsTOF Prosit 2023 model is available on Koina (Prosit_2023_intensity_timsTOF, <https://koina.proteomicsdb.org>) and can be used via Oktoberfest.

12. Not including peptides with free and cysteinylated cysteines is an important limitation of the current Prosit models for identifying immunopeptides. Similarly, dropping Cys-containing peptides during rescoring is an important limitation of the workflow presented in this study. Indeed, it has been shown that such peptides can represent an important part of the HLA-ligandome (see, e.g., ref. 39; and Bauza-Martinez, 2021). Although the authors mention it in the results, I think it is worth briefly mentioning this limitation in the discussion (e.g., in the second paragraph).

We agree with the reviewer that this is an important limitation. To highlight this more, we have added it to the discussion as well:

A limitation to take into consideration is that our model is not able to predict fragment ion intensities for peptides containing free cysteine side chains or PTMs that were not seen during training.

13. L260. the authors may mean that "there is a bias [in function] of the C-terminal arginine or lysine residues" and not "towards" them. See also my comment in the Results section (Comment 8). Please check and correct if applicable.

The sentence was adjusted accordingly:

This enabled the model to generalize over different peptide types, whereas machine learning models that are solely trained on tryptic data often fail to do so, for example, by exhibiting **strong biases based on the presence of C-terminal arginine or lysine residues.**

14. L276. Reference 40 was also trained on tryptic peptides and phosphopeptides for CCS prediction. It's worth mentioning it here for clarity.

The sentence was adjusted accordingly:

Some CCS prediction models already exist ^{22,39-41}, including a recent model trained on tryptic peptides, phosphopeptides, and immunopeptides ⁴⁰.

15. L277-285. I still think that this paragraph seems to indicate using collision energy calibration as a perspective and not a result. However, the authors added the explanation of the collision energy calibration to the Results section and highlighted its importance in the rebuttal document. In addition, the Oktoberfest manuscript mentions that it performs collision energy calibration. Thus, the authors may consider mentioning it at the end of the paragraph. For instance: "Similarly to what has been observed for retention time alignment [42], we expect a benefit from collision energy alignment to account for the run- to-run fluctuations. Therefore, the collision energy calibration is implemented in Oktoberfest for PSM rescoring [26]."

We followed the suggestion by the reviewer and the paragraph was adjusted accordingly:

Therefore, the collision energy calibration is implemented in Oktoberfest for PSM rescoring ²⁶.

Data availability

16. I downloaded again some of the files from the MassIVE dataset MSV000092456. When I unzip them, the raw folder (.d) is still borrowed in a series of folders. However, the authors said they corrected this. I may be accessing an old version, but the authors should make sure this is corrected for their final submission. For instance, "[my_folder]\Users\adams\Projects\non-tryptic-raw\HLAI_1_96_p2-H5_S2-H5_1_7014.d".

We thank the reviewer for bringing this to our attention. We were able to reproduce the same issue, so it was not due to accessing an old version. We contacted people from MassIVE and were able to resolve the issue. In addition, we double checked and downloaded some files and were able to confirm the files were corrected.

Multiple proteases experiment

Thank you for testing the performance of the timsTOF Prosit model on multiple proteases. It adds considerable value to the manuscript since it shows that the model can also be used for deep proteomics profiling. Thus, I have a few suggestions to highlight this further. However, I would understand if the authors prefer not to implement these and focus only on telling the story about improving immunopeptidomics experiments.

17. I suggest also mentioning the improvement in identifying peptides derived from proteomics experiments when the results are summarized in the Abstract, Introduction (last paragraph), and Discussion (first paragraph).

The section on deep proteomics profiling at the end of the second paragraph in the discussion was moved to right after the first paragraph:

In addition to the analysis of immunopeptidomics data, our model holds promise for numerous other biological and biomedical applications. One such area is deep proteome sequencing, where multiple proteases are used to enhance proteomic coverage³⁷, particularly in regions with suboptimal trypsin cleavage sites, such as membrane-spanning domains and splice junctions. Our model can effectively enhance the confidence of peptide identifications in such studies, enabling valuable insights into alternative splicing and facilitating a comprehensive exploration of its impact on the proteome.

After careful consideration, we have decided to not include these results in the abstract and introduction, to keep the main focus of the manuscript on improving immunopeptidomics experiments.

18. L177-186. To facilitate finding these results when reading the subtitles, the results could be presented in an additional subsection, independently of the immunopeptidomics experiments. In addition, Fig. S7b could be included in the main text instead of being borrowed.

The result section was moved to the end of the results and was given a subtitle:

PSM rescoring improves the identification rate of samples cleaved with different proteases

In addition, Supplementary Figure S7 was moved to the main manuscript as Figure 5.

Reviewer 3

The authors have adequately addressed the comments I made in a previous review. I recommend that the authors consider a few points for improvement.

We appreciate the reviewer's feedback and are glad to hear that the previous comments have been addressed. We carefully considered their suggestions for further improvement.

Comments

1. The timsTOF and HCD spectra appear to be quite different (Figure 1). Nevertheless, it seems that there were no differences between using timsTOF and HCD models in rescoring HLA-II peptides from timsTOF data. Could the authors add some comments on the reason for this?

Using the timsTOF Prosit 2023 model instead of the HCD Prosit 2020 model for PSM rescoring of HLA-II PSMs yields a small but consistent benefit. Both HCD and timsTOF have a beam-type fragmentation, making them more similar to each other compared to CID. This similarity could explain the relatively good performance of the HCD Prosit 2020 model on timsTOF data. It is important to note that other features, such as the number of theoretical ions observed in the experimental spectrum but not found in the predicted spectrum, can compensate for features based on differences in fragment ion intensity. Because HLA-II peptides are easier to identify, there are fewer PSMs left to rescue through PSM rescoring. Consequently, PSM rescoring with either model results in the identification of almost all true positive PSMs, leaving limited room for further improvement. This indicates that using any fragment ion intensity information, regardless of the model used, will result in an improved identification rate.

2. What is the difference between the data used in Figure 3 and Supplementary Figure S5? It is difficult to see the difference in the data. In addition, writing the numbers of PSMs in the legend is complicated (too many numbers).

The difference is that in Figure 3 the y-axis shows the number of unique peptides, compared to Figure S5 where the y-axis shows the number of PSMs. We included the values in the legend to show how many lost unique peptides or PSMs were present. However, we do agree that this makes the legend complicated. We solved this by adding a source data file containing all the values from grained, shared, and lost plots. We reference to the source data file in all relevant figures:

Source data are provided as a source data file.

3. Wasn't the data used in Supplementary Figure S6b Orbitrap-HCD? Yes, but it was referred to as timsTOF data in the legend. Also, please explicitly state Orbitrap CID for Figure S6a.

The legend incorrectly stated that in Figure S6b timsTOF HLA-II data is shown. We have corrected the legend accordingly. In addition, we have specified whether Orbitrap HCD or Orbitrap CID data is shown.

Supplementary Figure S6. Gained, shared, and lost peptide identifications for PSM rescoring with different fragment ion intensity prediction models. In general PSM rescoring was able to boost the number of PSMs, retaining true PSMs, gaining new PSMs, and losing only a small number of previously incorrect PSMs. **a** To evaluate the effect of the fragment ion intensity prediction model on PSM rescoring, the RT prediction-based features were excluded. On average PSM rescoring of HLA-I Orbitrap CID data with the timsTOF Prosit 2023 model resulted in a 2.0-fold increase. On average PSM rescoring of HLA-I timsTOF data with the HCD Prosit 2020 model resulted in a 1.9-fold increase. On average PSM rescoring of HLA-I timsTOF data with the CID Prosit 2020 model resulted in a 2.3-fold increase. **b** To evaluate the effect of the fragment ion intensity prediction model on PSM rescoring, the RT prediction-based features were excluded. On average PSM rescoring of HLA-II Orbitrap HCD data with the timsTOF Prosit 2023 model resulted in a 1.4-fold increase. On average PSM rescoring of HLA-II Orbitrap HCD data with the HCD Prosit 2020 model resulted in a 1.4-fold increase. On average PSM rescoring of HLA-II Orbitrap HCD data with the CID Prosit 2020 model resulted in a 1.3-fold increase. RCC = renal cell carcinoma; HNSCC = head and neck squamous-cell carcinoma; PBMC = peripheral blood mononuclear cell; CLL = chronic lymphocytic leukemia. Source data are provided as a source data file.